# TRAINING TENSOR ATTENTION EFFICIENTLY: FROM CUBIC TO ALMOST LINEAR TIME

## ABSTRACT

Tensor Attention, a multi-view attention that is able to capture high-order correlations among multiple modalities, can overcome the representational limitations of classical matrix attention. However, the $O(n^3)$ time complexity of tensor attention poses a significant obstacle to its utilization in transformers, where $n$ is the input sequence length. In this work, we prove that the backward gradient of tensor attention training can be computed in almost linear time $n^{1+o(1)}$, the same complexity as its forward computation under the bounded entries assumption. We provide a closed-form solution for the gradient and propose a fast computation method utilizing polynomial approximation methods and tensor algebraic techniques. Furthermore, we prove the necessity and tightness of our assumption through hardness analysis, showing that slightly weakening it renders the gradient problem unsolvable in truly subcubic time. Our theoretical results establish the feasibility of efficient higher-order transformer training and may facilitate practical applications of tensor attention architectures.

## 1 INTRODUCTION

The generative large language models (LLMs), such as Mistral 3.1 (Mistral, 2025), Llama 4 (Meta, 2025), Gemma 3 (Team et al., 2025), GPT-4o (OpenAI, 2025), Claude 3.7 (Anthropic, 2025), Grok 3 (xAI, 2025), Qwen 3 (Qwen, 2025), DeepSeek R1 (Guo et al., 2025), and many more have been widely involved in people's living and work in these two years, such as bio-informatics (Thirunavukarasu et al., 2023), coding (Hou et al., 2024), education (Kasneci et al., 2023), finance (Li et al., 2023b), law (Sun, 2023), and even writing NeurIPS conference reviews (Liang et al., 2024). The success of LLMs is based on the transformer architecture introduced by (Vaswani et al., 2017), which also has been introduced into other modalities (Dosovitskiy et al., 2020), such as vision-language models, e.g., CLIP (Radford et al., 2021), Flamingo (Alayrac et al., 2022), LLaMA-Adapter (Zhang et al., 2023a; Gao et al., 2023), LLava (Liu et al., 2024; 2023b), BLIP (Li et al., 2022; 2023a), MiniGPT-4 (Zhu et al., 2023), Qwen (Bai et al., 2023a;b), Gemini (Team et al., 2023), MM1 (McKinzie et al., 2024).

The above open-sourced large models use two-view matrix attention, i.e., each attention score/entry is related to two tokens (one query token and one key token) to capture the data correlation. More specifically, let $Z$ be hidden representations and $Q = ZW_Q, K = ZW_K, V = ZW_V$ be the corresponding query, key, and value matrices after projections using weights $W_Q, W_K, W_V$, respectively. Then, the classical/matrix attention head can be written as $\mathsf{Att}(Z) = \mathsf{Softmax}(QK^\top)V$.

On the other hand, many studies find that multi-view is crucial for high-order correlation in various kinds of data, e.g., math (Sanford et al., 2023), graph (Demirel et al., 2022; Luo et al., 2023), and multi-modality (Lahat et al., 2015). For example, recently, OpenAI released GPT-4o (OpenAI, 2025), and Google released Project Astra (Google, 2024), two flagship multi-modality models that aim to reason across three views, i.e., audio, vision, and text in real-time.

However, (Sanford et al., 2023) theoretically and empirically shows that classical attention can capture pairwise correlation but not triple-wise correlation due to the representational limitations of matrix attention. (Sanford et al., 2023) introduces a triple detection task, demonstrating that classical attention layers have complexity scaling linearly with the input size, while high-order attention can efficiently perform this computation within a single head.

In other words, one classical matrix attention head "cannot" capture the information relevant to multi-views simultaneously unless using multiple layers with careful architecture design. This poses a fundamental technical obstacle for multi-modality models to efficiently fuse multiple representations/views to capture the high-order correlation among them, e.g., the high-order correlations among multi-modalities such as audio, text, and images.

Table 1: Comparison to previous works.

| Reference | For/Backward | Matrix/Tensor |
|---|---|---|
| (Zandieh et al., 2023) | Forward | Matrix |
| (Alman & Song, 2023) | Forward | Matrix |
| (Han et al., 2024) | Forward | Matrix |
| (Alman & Song, 2024a) | Backward | Matrix |
| (Alman & Song, 2024b) | Forward | Tensor |
| Ours (Theorem 4.2) | Backward | Tensor |

To fundamentally solve the above obstacle, (Sanford et al., 2023) and (Alman & Song, 2024b) introduce *Tensor Attention*, which is a higher-order generalization of matrix attention that can capture high-order/multi-view information intrinsically. More specifically, it is defined as $\mathsf{Softmax}(Q(K_1 \oslash K_2)^\top)(V_1 \oslash V_2)$ (Definition 2.5, and illustrated in Figure 1), where $\oslash$ is column-wise Kronecker product, and $Q$, $K_1/V_1$, $K_2/V_2$ can be from different views/modalities. However, to implement Tensor Attention practically, we must overcome the complexity bottleneck. Let the input token length be $n$, then the forward and backward time complexity of tensor attention will be $O(n^3)$ as $Q(K_1 \oslash K_2)^\top \in \mathbb{R}^{n \times n^2}$ (Ma et al., 2019), while the time complexity of matrix attention is $O(n^2)$ only as $QK^\top \in \mathbb{R}^{n \times n}$ (Keles et al., 2023). For example, the input length of Llama2 (Touvron et al., 2023) is 4096. So, intuitively, if we put tensor attention in Llama2, the input length will reduce to 256 to keep the same complexity in running speed and memory consumption.

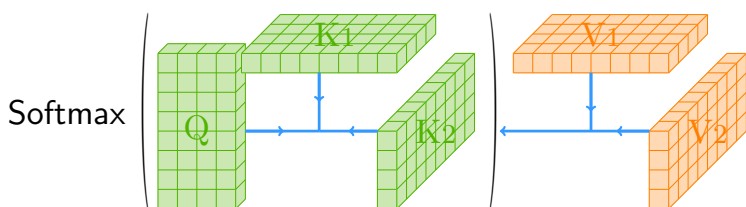

Figure 1: The visualization of tensor attention with $\mathsf{Softmax}$ activation function (Definition 2.5). We give an example of input token length $n = 8$, feature dimension $d = 4$.

There are several recent works to overcome the time complexity bottleneck above, e.g., $O(n^2)$ for matrix attention and $O(n^3)$ for tensor attention. (Zandieh et al., 2023) accelerate matrix attention forward via kernel density estimation and get truly sub-quadratic time running time. (Alman & Song, 2023) uses the polynomial approximation method to map the matrix attention into low-rank matrices during forward computation, leading to an almost linear time complexity $n^{1+o(1)}$ when entries are bounded. Similarly, under sparsity assumptions, (Han et al., 2024) achieves nearly linear time computation for matrix attention forward by identifying the larger entries in the attention matrix. On the one hand, with fine-grained analysis, (Alman & Song, 2024a) proposes a new backward algorithm to compute the gradient of matrix attention in almost linear time complexity $n^{1+o(1)}$ as well, under the same bounded entry assumption. On the other hand, (Alman & Song, 2024b) surprisingly finds that the *forward* computation of tensor attention can also be achieved in almost linear time $n^{1+o(1)}$ rather than almost quadratic time $n^{2+o(1)}$, under similar assumptions as (Alman & Song, 2023). See a summary in Table 1. Thus, it is natural to ask,

*Can we achieve almost linear time for gradient computation in Tensor Attention Training?*

We provide a positive answer in this work. Under the same bounded entries assumption as (Alman & Song, 2024b), we propose Algorithm 1 to fast compute the backward gradient of Tensor Attention Training in almost linear time $n^{1+o(1)}$ as its forward computation. Thus, our results may make the

tensor attention practical, as we can get around the $O(n^3)$ complexity barrier both in its forward and backward computation. **Our contributions are summarized as follows:**

- Under fine-grained analysis, we give the closed-form solution of the gradient computation of tensor attention (Lemma 3.1) and its time complexity without acceleration (Theorem 3.3).

- Based on the closed-form solution, by utilizing polynomial approximation methods and tensor computation techniques, we propose Algorithm 1 to fast compute the backward gradient of tensor attention training in almost linear time as its forward computation (Theorem 4.2).

- Furthermore, we prove that our assumption is necessary and "tight" by hardness analysis, i.e., if we slightly weaken the assumption, there is no algorithm that can solve the tensor attention gradient computation in truly sub-cubic complexity (Theorem 5.3).

**Roadmap.** In Section 2, we introduce the notations, several useful definitions, and our loss function. In Section 3, we give the closed form of the gradient of our loss function, and also its computational time complexity. In Section 4, we prove that we can compute the gradient in almost linear time. In Section 5, we provide the hardness analysis. In Section 6, we give the conclusion of our paper.

## 2 PRELIMINARY

In this section, we first provide the notations we use. In Section 2.1, we provide general definitions related to tensor operation. In Section 2.2, we provide key definitions that we utilize in this paper.

**Basic notations.** We use $[n]$ to denote $\{1, 2, \ldots, n\}$. We use $e_i$ to denote a column vector where only $i$-th location is 1 and zeros everywhere else. We denote an all 1 vector using $\mathbf{1}_n \in \mathbb{R}^n$. We use $\langle a, b \rangle$ to denote the inner product of $a, b \in \mathbb{R}^d$ i.e. $\langle a, b \rangle := \sum_{i=1}^d a_i b_i$. We use $\|x\|_p$ to denote the $\ell_p$ norm of a vector $x \in \mathbb{R}^n$, i.e. $\|x\|_2 := (\sum_{i=1}^n x_i^2)^{1/2}$, and $\|x\|_\infty := \max_{i \in [n]} |x_i|$. We use $\circ$ to denote the Hadamard product, i.e., the $(i, j)$-entry of $A \circ B$ is $A_{i,j} B_{i,j}$. We use $\mathrm{tr}[A] := \sum_{i=1}^n A_{i,i}$ to denote the trace of a matrix $A \in \mathbb{R}^{n \times n}$. We use $\exp(A)$ to denote a matrix where $\exp(A)_{i,j} := \exp(A_{i,j})$ for a matrix $A \in \mathbb{R}^{n \times d}$. We use $\|A\|_\infty$ to denote the $\ell_\infty$ norm of a matrix $A \in \mathbb{R}^{n \times d}$, i.e. $\|A\|_\infty := \max_{i \in [n], j \in [d]} |A_{i,j}|$. We use $\|A\|_F$ to denote the Frobenius norm of a matrix $A \in \mathbb{R}^{n \times d}$, i.e. $\|A\|_F := \sqrt{\sum_{i \in [n]} \sum_{j \in [d]} |A_{i,j}|^2}$. We use $\mathrm{poly}(n)$ to denote polynomial time complexity w.r.t. $n$.

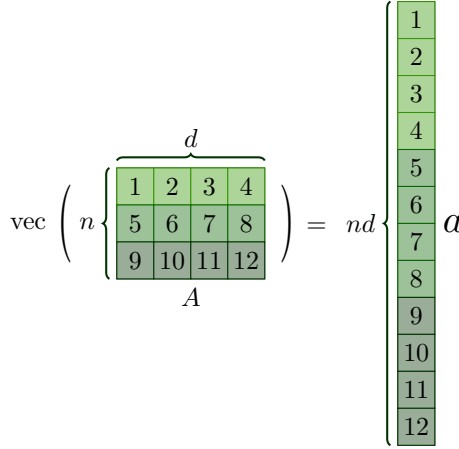

Figure 2: The visualization of vectorization operator $\mathrm{vec}(\cdot)$, which stacks rows of a matrix $A \in \mathbb{R}^{n \times d}$ into a column vector $a \in \mathbb{R}^{nd}$. In this figure, we give an example of $n = 3, d = 4$. The components of $A$ and $a$ are also given for easier understanding.

**Tensor related notations.** Let $A \in \mathbb{R}^{n \times d}$. We use $a := \mathrm{vec}(A)$ to denote the length $nd$ vector obtained by stacking rows of $A$ into a column vector, i.e. $\mathrm{vec}(A) := [a_1^\top, a_2^\top, \ldots, a_n^\top]^\top$ where $a_i^\top$ is the $i$-th row of $A$, or simply $\mathrm{vec}(A)_{j+(i-1)d} := A_{i,j}$ for any $i \in [n], j \in [d]$, visualized in Fig. 2. Let $I_d \in \mathbb{R}^{d \times d}$ denote the identity matrix. Let $\mathsf{I}_d \in \mathbb{R}^{d \times d \times d}$ denote the identity tensor, i.e., the diagonal entries are 1 and zeros everywhere else. Let $X \in \mathbb{R}^{d \times d^2}$. Let $x \in \mathbb{R}^{d^3}$ denote the vectorization of $X \in \mathbb{R}^{d \times d^2}$. Let $\mathsf{X} \in \mathbb{R}^{d \times d \times d}$ be the tensorization of $X \in \mathbb{R}^{d \times d^2}$, where $\mathsf{X}_{a,b,c} = X_{a,(b-1)d+c}$ for any $a, b, c \in [d]$. We define the corresponding function $\mathrm{mat} : \mathbb{R}^{d \times d \times d} \to \mathbb{R}^{d \times d^2}$ as $X = \mathrm{mat}(\mathsf{X})$ where $X_{a,(b-1)d+c} = \mathsf{X}_{a,b,c}$ for any $a, b, c \in [d]$.

## 2.1 Definition of Tensor Operations

We define some operations like the Kronecker product, which is a matrix operation applied to two matrices of any size, producing a block matrix. It is different from regular matrix multiplication and will be useful for our introduction and analysis of tensor attention.

**Definition 2.1** ($\otimes$ Kronecker product). *Given $K_1 \in \mathbb{R}^{n_1 \times d_1}$ and $K_2 \in \mathbb{R}^{n_2 \times d_2}$, for any $i_1 \in [n_1], i_2 \in [n_2], j_1 \in [d_1], j_2 \in [d_2]$, we define the matrix $K := K_1 \otimes K_2 \in \mathbb{R}^{n_1 n_2 \times d_1 d_2}$ as*

$$K_{i_1 + (i_2-1)n_1, j_1 + (j_2-1)d_1} = (K_1)_{i_1, j_1} \cdot (K_2)_{i_2, j_2}.$$

In this work, we will primarily use the following column-wise and row-wise versions of the Kronecker product, which are special kinds of Kronecker products.

**Definition 2.2** ($\oslash$ column-wise Kronecker product, also known as Kathri-Rao product). *Given matrices $K_1 \in \mathbb{R}^{n_1 \times d}, K_2 \in \mathbb{R}^{n_2 \times d}$, we define the matrix $K := K_1 \oslash K_2 \in \mathbb{R}^{n_1 n_2 \times d}$ as follows*

$$K_{i_1 + (i_2-1)n_1, j} := (K_1)_{i_1, j} \cdot (K_2)_{i_2, j}, \quad \forall i_1 \in [n_1], i_2 \in [n_2], j \in [d].$$

**Definition 2.3** ($\ominus$ row-wise Kronecker product, also referred to as the face-splitting product). *Given matrices $K_1 \in \mathbb{R}^{n \times d_1}, K_2 \in \mathbb{R}^{n \times d_2}$, we define the matrix $K := K_1 \ominus K_2 \in \mathbb{R}^{n \times d_1 d_2}$ as follows*

$$K_{i, j_1 + (j_2-1)d_1} := (K_1)_{i, j_1} \cdot (K_2)_{i, j_2}, \quad \forall i \in [n], j_1 \in [d_1], j_2 \in [d_2].$$

## 2.2 Key Definitions of Tensor Attention

Now, we are ready to introduce the tensor attention. First, we introduce the parameters and input.

**Definition 2.4** (Input and weight matrix). *We define the input sequence as $Z \in \mathbb{R}^{n \times d}$ and the key, query, and value weight matrix as $W_{K_1}, W_{K_2}, W_Q, W_{V_1}, W_{V_2} \in \mathbb{R}^{d \times d}$. Then, we define the key, query, and value matrix as $K_1 := ZW_{K_1} \in \mathbb{R}^{n \times d}, K_2 := ZW_{K_2} \in \mathbb{R}^{n \times d}, Q := ZW_Q \in \mathbb{R}^{n \times d}, V_1 := ZW_{V_1} \in \mathbb{R}^{n \times d}, V_2 := ZW_{V_2} \in \mathbb{R}^{n \times d}$.*

Then, based on the Kronecker product, we define tensor attention in the following way.

**Definition 2.5** (Tensor attention, Definition 7 in (Sanford et al., 2023), Definition 1.1 in (Alman & Song, 2024b)). *Given input matrices $Q, K_1, K_2, V_1, V_2 \in \mathbb{R}^{n \times d}$, compute the matrix*

$$\underbrace{D^{-1}}_{n \times n} \underbrace{A}_{n \times n^2} \underbrace{V}_{n^2 \times d} \in \mathbb{R}^{n \times d},$$

*where (1) $A := \exp(QK^\top / d) \in \mathbb{R}^{n \times n^2}$ and $K := K_1 \oslash K_2 \in \mathbb{R}^{n^2 \times d}$, (2) $D := \mathrm{diag}(A \mathbf{1}_{n^2}) \in \mathbb{R}^{n \times n}$, and (3) $V := V_1 \oslash V_2 \in \mathbb{R}^{n^2 \times d}$.*

**Remark 2.6.** *In Definition 2.5, on the one hand, we separate the Softmax operation into an element-wise $\exp$ operation and a diagonal normalization matrix $D$ for a more transparent formulation. On the other hand, we change $K, V \in \mathbb{R}^{n \times d}$ in classical attention to $K_1 \oslash K_2, V_1 \oslash V_2 \in \mathbb{R}^{n^2 \times d}$ in tensor attention, where $\oslash$ is column-wise Kronecker product defined in Definition 2.2.*

Our Definition 2.5 covers the self-attention setting when the query/key/values $Q, K_1, K_2, V_1, V_2$ follow Definition 2.4 where they share the same input. It is then a tensor self-attention, which can capture high-order information of the input $Z$. When the query/key/values have different inputs, it is then a tensor cross-attention that can capture high-order relationships among multiple inputs.

Also, note that we have $A \in \mathbb{R}^{n \times n^2}$ in Definition 2.5. Although $QK^\top$ is a low-rank matrix with rank at most $d$, $\exp(QK^\top)$ may be a full-rank matrix in general. Thus, it is clear to see the exact forward computation of tensor attention takes $O(n^3)$ time. Here, we introduce a forward tensor attention approximation task, which will help us formulate the tensor attention gradient approximation task later. Furthermore, (Alman & Song, 2024b) show that they can solve this approximation task in almost linear time $n^{1+o(1)}$ (Lemma 4.1).

**Definition 2.7** (Approximate Tensor Attention Computation (ATAttC$(n, d, B, \epsilon)$), Definition 1.2 in (Alman & Song, 2024b)). *Given input matrices $Q, K_1, K_2, V_1, V_2 \in \mathbb{R}^{n \times d}$ and parameters $\epsilon, B > 0$, where $\max\{\|Q\|_\infty, \|K_1\|_\infty, \|K_2\|_\infty, \|V_1\|_\infty, \|V_2\|_\infty\} \leq B$. Let $A, D, V$ be defined in Definition 2.5. Then, our target is to output a matrix $T \in \mathbb{R}^{n \times d}$ satisfying*

$$\|\underbrace{T}_{n \times d} - \underbrace{D^{-1}}_{n \times n} \underbrace{A}_{n \times n^2} \underbrace{V}_{n^2 \times d}\|_\infty \leq \epsilon.$$

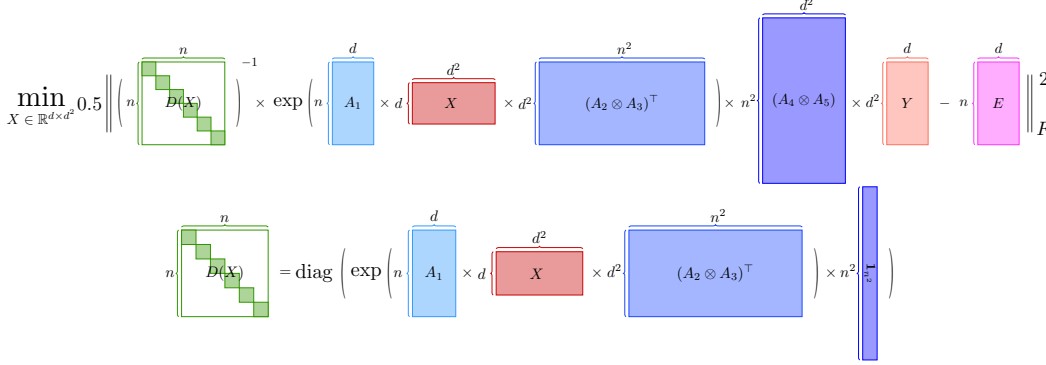

Figure 3: The visualization of loss function defined in Definition 2.8. Let $A_1, A_2, A_3, A_4, A_5$ and $E$ be $n \times d$ input matrices. Let $Y$ be a given matrix with size $d^2 \times d$. The Kronecker product operator $\otimes$ is defined in Definition 2.1. We minimize matrix $X \in \mathbb{R}^{d \times d^2}$ in our loss function. We first compute $\exp(A_1 X (A_2 \otimes A_3)^\top)$. Then, we compute $D(X) := \mathrm{diag}(\exp(A_1 X (A_2 \otimes A_3)^\top) \mathbf{1}_{n^2})$. Afterwards, we compute $D(X)^{-1} \exp(A_1 X (A_2 \otimes A_3)^\top)(A_4 \otimes A_5)Y - E$. Finally, we optimize $X$ to compute the minimum of its Frobenius norm with a scaling factor $0.5$.

For our focus, tensor attention training, we would like to find weights to fit the tensor attention to a desired output $E$. We first simplify the attention expression of Definition 2.5, whose inputs are from Definition 2.4 with weight matrices $W_Q, W_{K_1}, W_{K_2}, W_{V_1}, W_{V_2} \in \mathbb{R}^{d \times d}$. Let $X := W_Q \cdot (W_{K_1} \oslash W_{K_2})^\top \in \mathbb{R}^{d \times d^2}$ and $Y := W_{V_1} \oslash W_{V_2} \in \mathbb{R}^{d^2 \times d}$. It can be verified that the tensor attention equals

$$D^{-1} \exp(ZX(Z \otimes Z)^\top / d)(Z \otimes Z)Y,$$

where $Z \in \mathbb{R}^{n \times d}$ is defined as the input sequence in Definition 2.4.

The naive gradient computation for the tensor attention training takes $\Omega(n^3)$ time. The gradient for $X$ is the bottleneck while that for $Y$ is not, since $A_1 X (A_2 \otimes A_3)^\top \in \mathbb{R}^{n \times n^2}$ lies in the non-linear function Softmax. Also, note that with gradients of $X$ and $Y$, it is easy to get the gradients of the weight matrices $W_Q, W_{K_1}, W_{K_2}, W_{V_1}, W_{V_2}$. Therefore, we model the tensor attention training as the following tensor attention optimization problem (where $A_1, A_2, A_3, A_4, A_5$ are introduced to replace $Z$ to capture more general settings such as cross-attention). See Figure 3 for an illustration.

**Definition 2.8** (Tensor attention optimization). *Suppose that $A_1, A_2, A_3, A_4, A_5, E \in \mathbb{R}^{n \times d}$ and $Y_1, Y_2 \in \mathbb{R}^{d \times d}$ are given. We formulate the attention optimization problem*

$$\min_{X \in \mathbb{R}^{d \times d^2}} \mathsf{Loss}(X)$$

*as*

$$0.5 \| D(X)^{-1} \exp(A_1 X (A_2 \otimes A_3)^\top / d)(A_4 \otimes A_5)Y - E \|_F^2,$$

*where (1) $A_2 \otimes A_3 \in \mathbb{R}^{n^2 \times d^2}$ is the tensor product between $A_2$ and $A_3$, (2) $D(X) = \mathrm{diag}(\exp(A_1 X (A_2 \otimes A_3)^\top / d) \mathbf{1}_{n^2}) \in \mathbb{R}^{n \times n}$, and (3) $Y = Y_1 \oslash Y_2 \in \mathbb{R}^{d^2 \times d}$.*

Our main focus is the following Approximate Tensor Attention Loss Gradient Computation task.

**Definition 2.9** (Approximate Tensor Attention Loss Gradient Computation (ATAttLGC$(n, d, B, \epsilon)$)). *Let $\epsilon, B > 0$. Let $A_1, A_2, A_3, A_4, A_5, E \in \mathbb{R}^{n \times d}$ and let $X_1, X_2, X_3, Y_1, Y_2 \in \mathbb{R}^{d \times d}$ (see Definition 2.8). Let $X = X_1 \cdot (X_2 \oslash X_3)^\top \in \mathbb{R}^{d \times d^2}$. Assume that $\max\{\|A_1 X_1\|_\infty, \|A_2 X_2\|_\infty, \|A_3 X_3\|_\infty, \|A_4 Y_1\|_\infty, \|A_5 Y_2\|_\infty\} \le B$. Let us assume that any numbers in the previous matrices are in the $\log(n)$ bits model[1]. We define $\mathsf{Loss}(X)$ the*

---

[1]Each entry in the matrix is represented by at most $\log(n)$ bits. This assumption is well-accepted and widely used in the computational complexity community, e.g., (Feng et al., 2024; Liu et al., 2023a; Merrill & Sabharwal, 2023).

same as Definition 2.8. Let the gradient of loss function $\mathsf{Loss}(X)$ be $\frac{\mathrm{dLoss}(X)}{\mathrm{d}X} \in \mathbb{R}^{d \times d^2}$. Then, our target is to output a matrix $\widetilde{g} \in \mathbb{R}^{d \times d^2}$ satisfying

$$\|\widetilde{g} - \frac{\mathrm{dLoss}(X)}{\mathrm{d}X}\|_\infty \leq \epsilon.$$

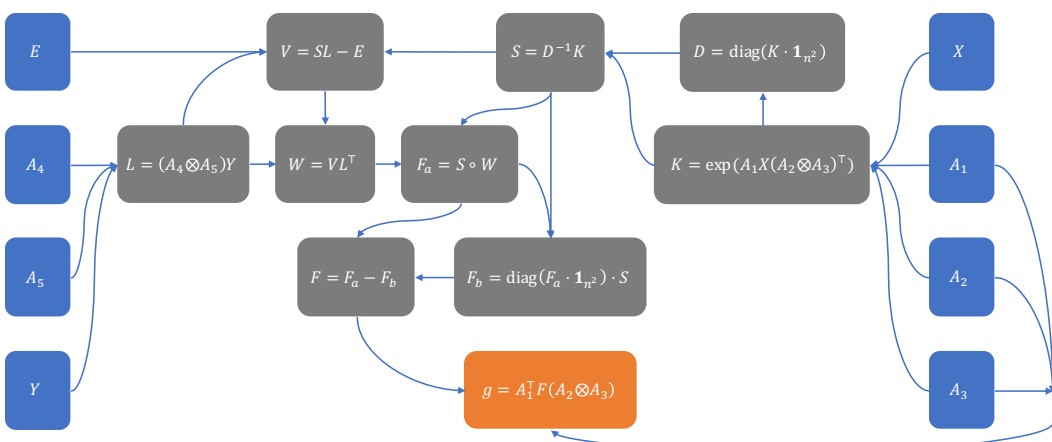

Figure 4: The computational graph for tensor attention backward. The blue boxes are input matrices, the gray boxes are intermediate matrices, and the orange box is the final gradient matrix. Here, $A_1, A_2, A_3, A_4, A_5$ denote the previous inputs, $E$ denotes the target matrix, and $X, Y$ denote the attention weights. More detailed definitions of each variable can be found in Section D, E and F.

## 3 EXACT TENSOR ATTENTION GRADIENT COMPUTATION AND COMPLEXITY

In this section, we provide the closed form of the tensor attention gradient of the loss function (Definition 2.8) and also its computational time. First, we calculate the closed form of the gradient in the following lemma, whose proof is in Appendix E.5.

**Lemma 3.1** (Closed form of gradient, informal version of Lemma E.6). *Define the function $\mathsf{F}(x) \in \mathbb{R}^{n \times n^2}$ as in Definition D.6 (see Fig. 4 for an illustration). Suppose that $A_1, A_2, A_3 \in \mathbb{R}^{n \times d}$ are three given matrices. Suppose that $\mathsf{Loss}(x)$ is defined as Definition 2.8, where $x = \mathrm{vec}(X)$. Then, we have*

$$\frac{\mathrm{dLoss}(x)}{\mathrm{d}x} = \mathrm{vec}(A_1^\top \mathsf{F}(x)(A_2 \otimes A_3)) \in \mathbb{R}^{d^3}.$$

Note that $\mathsf{F}(x)$ is a size $n \times n^2$ matrix which is the bottleneck obstacle in time complexity.

**Definition 3.2.** *Let $\mathcal{T}_{\mathrm{mat}}(a, b, c)$ denote the time of multiplying $a \times b$ matrix and $b \times c$ matrix.*

Then, with straightforward analysis, we get the following theorem about the time complexity of naive computation. The complete proof is in Appendix E.6.

**Theorem 3.3** (Tensor attention gradient computation, informal version of Theorem E.7). *Suppose that $A_1, A_2, A_3, A_4, A_5, E \in \mathbb{R}^{n \times d}$ are input fixed matrices. We denote matrix variables as $X \in \mathbb{R}^{d \times d^2}$ and $Y \in \mathbb{R}^{d^2 \times d}$ (gradient computation is w.r.t. $X$). Let $g = \frac{\mathrm{dLoss}(X)}{\mathrm{d}X} \in \mathbb{R}^{d \times d^2}$ (for definition of $\mathsf{Loss}(X)$, see Definition 2.8). Then, we show that computing the gradient $g \in \mathbb{R}^{d \times d^2}$ requires $\mathcal{T}_{\mathrm{mat}}(n, d^2, n^2)$ time.*

Note that $\mathcal{T}_{\mathrm{mat}}(n, d^2, n^2) \geq \Omega(n^3)$. Thus, the naive tensor attention gradient computation is a complexity obstacle in practice, as discussed in Section 1. Based on the closed formulation in Lemma 3.1, we derive our acceleration method, which will be introduced in the following section.

# 4 FAST TENSOR ATTENTION GRADIENT COMPUTATION

In this section, we show how to compute the tensor attention matrix gradient in almost linear time. In Section 4.1, we demonstrate our main results. In Section 4.2, we introduce some key tensor techniques used in our proof.

---

**Algorithm 1** Almost Linear Time Tensor Attention Gradient Computation

---

1: **procedure** FASTTENSORATTENTION($A_1, A_2, A_3, A_4, A_5, E \in \mathbb{R}^{n \times d}, X_1, X_2, X_3, Y_1, Y_2 \in \mathbb{R}^{d \times d}, n \in \mathbb{N}_+, d \in \mathbb{N}_+, \epsilon \in (0, 0.1))$ ▷ Definition 2.9, Theorem 4.2
2: ▷ $n$ can be viewed as the length of the sentence
3: ▷ $d$ can be viewed as the feature of dimension, and we assume $d = O(\log n)$
4: ▷ $\epsilon$ is the accuracy output, and we typically pick $1/\operatorname{poly}(n)$
5:     Get $U_1, V_1, W_1 \in \mathbb{R}^{n \times n^{o(1)}}$ to approximate $\mathsf{S}(x)$ via Lemma F.1 ▷ $O(n^{1+o(1)})$ time
6:     $U_2 \leftarrow U_1(V_1 \oslash W_1)^\top \mathsf{L}(y) - E$ to approximate $\mathsf{V}(x)$ via Lemma 4.5 ▷ $O(n^{1+o(1)})$ time
7:     $V_2, W_2 \leftarrow A_4 Y_1, A_5 Y_2$ to approximate $\mathsf{W}(x)$ via Lemma F.3 ▷ $O(nd^2)$ time
8:     $U_3, V_3, W_3 \leftarrow U_1 \ominus U_2, V_1 \ominus V_2, W_1 \ominus W_2$ to approximate $\mathsf{F}_a(x)$ via Lemma F.5 ▷ $O(n^{1+o(1)})$ time
9:     Precompute $V_1^\top V_2$ and $W_1^\top W_2$ to approximate $\mathsf{F}_b(x)$ via Lemma F.7 ▷ $O(n^{1+o(1)})$ time
10:     **for** $j_0 \in [n]$ **do** ▷ Overall $\widetilde{\mathsf{R}}(x)$ takes $O(n^{1+o(1)})$ time
11:         $\widetilde{\mathsf{R}}(x)_{j_0} \leftarrow (U_1)_{j_0,*}((V_1^\top V_2) \circ (W_1^\top W_2))((U_2)_{j_0,*})^\top$
12:     **end for**
13:     $U_4 \leftarrow \operatorname{diag}(\widetilde{\mathsf{R}}(x))U_1$ ▷ $O(n^{1+o(1)})$ time
14:     $V_4, W_4 \leftarrow V_1, W_1$ ▷ $O(n^{1+o(1)})$ time
15:     /* Approximate $\mathsf{F}(x)$, Theorem F.8 */
16:     $U_5, V_5, W_5 \leftarrow [U_3, -U_4], [V_3, V_4], [W_3, W_4]$ ▷ $O(n^{1+o(1)})$ time
17:     /* Approximate $g$, Theorem F.8 */
18:     Precompute $A_1^\top U_5, A_2^\top V_5, A_3^\top W_5$ separately ▷ $O(dn^{1+o(1)})$ time
19:     $\widetilde{g} \leftarrow (A_1^\top U_5) \odot (A_2^\top V_5) \odot (A_3^\top W_5)$ ▷ $\odot$ in Definition C.3. $O(d^3 n^{o(1)})$ time
20:     **return** $\widetilde{g}$ ▷ As $d = O(\log n)$, the total complexity is $O(n^{1+o(1)})$ time
21: **end procedure**

---

## 4.1 MAIN RESULTS FOR FAST GRADIENT COMPUTATION

Polynomial approximation methods involve representing complex functions through simpler polynomial forms to facilitate easier analysis and computation. They are crucial in numerical analysis, aiding in the efficient solution of differential equations and optimization problems, and are widely used in simulations and machine learning (Aggarwal & Alman, 2022; Alman et al., 2020).

Based on the polynomial approximation methods, (Alman & Song, 2024b) get the following result about tensor attention acceleration, which will be used to prove our main result.

**Lemma 4.1** (Theorem 1.4 in (Alman & Song, 2024b))*. There is an algorithm that solves* $\mathsf{ATAttC}(n, d = O(\log n), B = o(\sqrt[3]{\log n}), \epsilon = 1/\operatorname{poly}(n))$ *(see Definition 2.7) in time* $n^{1+o(1)}$.

Using similar polynomial approximation methods, and combined with a series of tensor analysis techniques (Section 4.2), we get our main acceleration results.

**Theorem 4.2** (Main result for fast gradient computation, informal version of Theorem F.8)*. Assuming the entries of* $A_1, A_2, A_3, A_4, A_5, E \in \mathbb{R}^{n \times d}$ *and* $X_1, X_2, X_3, Y_1, Y_2 \in \mathbb{R}^{d \times d}$ *are represented using* $O(\log n)$ *bits. Then, there exist an algorithm (Algorithm 1) that runs in* $n^{1+o(1)}$ *time to solve* $\mathsf{ATAttLGC}(n, d = O(\log n), B = o(\sqrt[3]{\log n}), \epsilon = 1/\operatorname{poly}(n))$ *(see Definition 2.9), i.e., our algorithm computes a gradient matrix* $\widetilde{g} \in \mathbb{R}^{d \times d^2}$ *satisfying*

$$\|\frac{\mathrm{dLoss}(X)}{\mathrm{d}X} - \widetilde{g}\|_\infty \leq 1/\operatorname{poly}(n).$$

*Proof sketch of Theorem 4.2.* The complete proof can be found in Appendix F.6.

We use the polynomial approximation method to obtain low-rank approximation results for $D^{-1} \exp(A_1 X(A_2 \otimes A_3)^\top/d)$ in Lemma F.1. However, this cannot be directly used for the closed form of the tensor attention gradient solution in Theorem 3.3. Utilizing a series of tensor techniques (Section 4.2 and Appendix C), we smartly convey these low rank properties throughout the gradient formulation and computation, where two key steps are fixed in Lemma F.5 and Lemma F.7. $\square$

**Remark 4.3.** *The assumption in Theorem 4.2 is practical. In practice, especially in recent long context tasks, the $n$ is large, e.g., $n = 2 \times 10^6$ for Google's Gemini 1.5 Pro (Gemini, 2024), while the model training uses a half-precision floating-point format, e.g., the bit number is 16. Furthermore, our assumption is "tight", where if we slightly weaken the assumption, there is no algorithm that can solve the tensor attention gradient computation in truly sub-cubic complexity (Theorem 5.3).*

Our Theorem 4.2 accurately approximates ($\epsilon = 1/\mathrm{poly}(n)$) the tensor attention gradient computation in almost linear time $n^{1+o(1)}$ under practical assumptions (see the above Remark 4.3). Thus, our methods solve the last puzzle of tensor attention acceleration. Combined with previous work on tensor attention inference, this may make tensor attention practical, as we overcome the theoretical cubic time complexity barrier both in inference and training.

We provide Algorithm 1 for our almost linear time tensor attention training method. In the detailed algorithm, first, we construct $U_1, V_1, W_1$ in Lemma F.1. Then, we construct $U_2, V_2, W_2$ in Lemma F.3 and $U_3, V_3, W_3$ in Lemma F.5. We show how to construct $U_4, V_4, W_4$ in Lemma F.7. Finally, we construct $U_5, V_5, W_5$ and compute the gradient $g$ in almost linear time in Theorem F.8.

## 4.2 Tensor Operation Analysis Techniques

Here, we introduce some key techniques for proving Theorem 4.2. These techniques make it possible to convey the low-rank property even during the tensor operations, solving the novel technical challenges in tensor attention gradient computation.

We first introduce a distributed rule, where the proof is in Appendix C.2.

**Fact 4.4.** *Let $U_1 \in \mathbb{R}^{n_1 \times d}$ and $U_2 \in \mathbb{R}^{n_1 \times k}$. Let $V_1 \in \mathbb{R}^{n_2 \times d}$ and $V_2 \in \mathbb{R}^{n_2 \times k}$. Let $W_1 \in \mathbb{R}^{n_3 \times d}$ and $W_2 \in \mathbb{R}^{n_3 \times k}$. We have*

$$\underbrace{(U_1 \ominus U_2)}_{n_1 \times dk} \cdot (\underbrace{(V_1 \ominus V_2)}_{n_2 \times dk} \oslash \underbrace{(W_1 \ominus W_2)}_{n_3 \times dk}))^\top = (\underbrace{U_1}_{n_1 \times d} (\underbrace{V_1}_{n_2 \times d} \oslash \underbrace{W_1}_{n_3 \times d})^\top) \circ (\underbrace{U_2}_{n_1 \times k} (\underbrace{V_2}_{n_2 \times k} \oslash \underbrace{W_2}_{n_3 \times k})^\top)$$

Fact 4.4 tells us that the multiple tensor operation can be distributed to a different format. If we have some low-rank matrix/tensor, we can distribute them into each component so that each component can be accelerated via the low-rank property. Intuitively, this allows us to borrow some low-rank benefits from other terms to fix the bottleneck terms.

Then, we provide an important tool whose proof is in Appendix C.2.

**Lemma 4.5** (Informal version of Lemma C.13). *Given $A_1 \in \mathbb{R}^{n_1 \times d_1}$, $A_2 \in \mathbb{R}^{n_2 \times d_1}$, let $A := (A_1 \oslash A_2) \in \mathbb{R}^{n_1 n_2 \times d_1}$. Given $B_1 \in \mathbb{R}^{n_1 \times d_2}$, $B_2 \in \mathbb{R}^{n_2 \times d_2}$, let $B := (B_1 \oslash B_2) \in \mathbb{R}^{n_1 n_2 \times d_2}$. We define $C \in \mathbb{R}^{d_1 \times d_2}$ as $C := A^\top B$ and $C_1 := A_1^\top B_1 \in \mathbb{R}^{d_1 \times d_2}$, $C_2 := A_2^\top B_2 \in \mathbb{R}^{d_1 \times d_2}$. Then, we have $C_1 \circ C_2 = C$ and given $A_1, A_2, B_1, B_2$, we can get $C$ in $\mathcal{T}_{\mathrm{mat}}(d_1, \max\{n_1, n_2\}, d_2)$ time.*

Lemma 4.5 is a highly non-trivial method to handle tensor operation, $\circ$ and matrix multiplication together. By using the method, we save the computation time from $\mathcal{T}_{\mathrm{mat}}(d, n^2, d)$ to $\mathcal{T}_{\mathrm{mat}}(d, n, d)$, which gets rid of the bottleneck quadratic term $n^2$.

Lastly, we introduce a tensor trick, which can reduce a tensor operation to a matrix multiplication operation. The proof is in Appendix C.3.

**Fact 4.6** (Tensor-trick). *Given matrices $A_1 \in \mathbb{R}^{n_1 \times d_1}, A_2 \in \mathbb{R}^{n_2 \times d_2}$ and $X \in \mathbb{R}^{d_1 \times d_2}$, we have $\mathrm{vec}(A_1 X A_2^\top) = (A_1 \otimes A_2)\mathrm{vec}(X) \in \mathbb{R}^{n_1 n_2}$.*

## 5 Tensor Attention Gradient Complexity Lower Bound

In this section, we show that our assumption is necessary. First, we introduce some hardness analysis background in Section 5.1. Then, we introduce our main hardness result in Section 5.2.

## 5.1 SETH and Tensor Attention Forward Hardness

We provide the findings that our results are based on. We first introduce a well-known hypothesis in hardness analysis. The Strong Exponential Time Hypothesis (SETH), a well-established conjecture, has been instrumental in establishing fine-grained lower bounds for numerous algorithmic problems, as highlighted in the survey by (Williams, 2018). More than two decades ago, (Impagliazzo & Paturi, 2001) introduced SETH as an enhanced version of the well-known $\mathsf{P} \neq \mathsf{NP}$ conjecture, positing that current algorithms solving the SAT problem are nearly optimal in terms of efficiency.

**Hypothesis 5.1** (Strong Exponential Time Hypothesis (SETH), (Impagliazzo & Paturi, 2001))**.** *Given $\epsilon > 0$, there exists $k \geq 3 \in \mathbb{Z}$ such that it is impossible to solve $k$-SAT problem with $n$ variables in $O(2^{(1-\epsilon)n})$ time, including using any randomized algorithms.*

We will critically utilize the hardness result of the forward tensor attention computation.

**Lemma 5.2** (Theorem 1.3 in (Alman & Song, 2024b))**.** *Assuming SETH, for any constant $\delta > 0$, no algorithm can solve $\mathsf{ATAttC}(n, d = \Theta(\log n), B = \Theta(\sqrt[3]{(1+\gamma)\log n}), \epsilon = n^{\gamma-O(1)})$ (Definition 2.7) in $O(n^{3-\delta})$ time, even if the inputs meet the following conditions for any $\gamma \geq 0$: (1) $V \in \{0,1\}^{n^2 \times d}$, (2) There exists $B_a \leq O((1+\gamma)\log^2 n) = O(d(\sqrt[3]{(1+\gamma)\log n})^3)$ where all entries of $Q(K_1 \oslash K_2)^\top$ are within the range $[1, B_a]$ and more than half entries in each row of $Q(K_1 \oslash K_2)^\top$ are equal to $B_a$.*

This result shows that assuming SETH, if we just slightly weaken the assumption from $B = O(\sqrt[3]{\log n})$ to $B = \Theta(\sqrt[3]{(1+\gamma)\log n})$ with $\gamma = \omega(1)$, then the tensor attention forward computation is hard, i.e., no algorithm can solve it in truly sub-cubic time.

## 5.2 Main Result for Hardness

Based on the above observation (Lemma 5.2), we prove our main result for tensor attention gradient computation hardness.

**Theorem 5.3** (Main result for hardness, informal version of Theorem G.3)**.** *Let $\gamma : \mathbb{N} \to \mathbb{N}$ be any function with $\gamma(n) = o(\log n)$ and $\gamma(n) = \omega(1)$. Assuming SETH, for any constant $\delta > 0$, it is impossible to solve $\mathsf{ATAttLGC}(n, d = \Theta(\log n), B = \Theta(\sqrt[3]{\gamma(n) \cdot \log n}), \epsilon = O(1/(\log n)^4))$ (Definition 2.9) in time $O(n^{3-\delta})$ when $E = 0$, $\mathsf{Y} = \mathsf{I}_d$, $\mathsf{X} = \lambda \mathsf{I}_d$ for some scalar $\lambda \in [0, 1]$.*

See the formal proof in Appendix G.2. The intuition is that if we can solve $\mathsf{ATAttLGC}$ in $O(t)$ time, then we can solve $\mathsf{ATAttC}$ in $O(t \cdot \log^{11}(n))$ time by interpolation and "integral". We see a similar sharp complexity transition as forward computation (Lemma 5.2): assuming SETH, if we slightly weaken the assumption from $B = O(\sqrt[3]{\log n})$ to $B = \Theta(\sqrt[3]{(1+\gamma)\log n})$ with $\gamma = \omega(1)$, then the tensor attention gradient computation will be unsolvable in truly sub-cubic time as well.

## 6 Discussion and Conclusion

In this work, we proved that the backward gradient of tensor attention training can be computed in almost linear $n^{1+o(1)}$ time, the same complexity as its forward computation, under a bounded entries assumption. We provided a closed-form solution for the gradient and proposed a fast computation method utilizing polynomial approximation and tensor algebraic techniques. Furthermore, we proved the necessity and tightness of our assumption through hardness analysis, showing that slightly weakening it renders the tensor attention gradient problem unsolvable in truly subcubic time.

Our theoretical results establish the feasibility of efficient higher-order transformer training and may facilitate practical applications of tensor attention architectures. Due to space limits, we provide our further discussion and extension in Appendix A. Future work can perform empirical evaluations of the method in practical large language models, and explore how these findings can be implemented in real-world scenarios to enable the development of powerful higher-order models.

ETHIC STATEMENT

This paper does not involve human subjects, personally identifiable data, or sensitive applications. We do not foresee direct ethical risks. We follow the ICLR Code of Ethics and affirm that all aspects of this research comply with the principles of fairness, transparency, and integrity.

REPRODUCIBILITY STATEMENT

We ensure reproducibility of our theoretical results by including all formal assumptions, definitions, and complete proofs in the appendix. The main text states each theorem clearly and refers to the detailed proofs. No external data or software is required.

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

# Appendix

**Roadmap.** In Section A, we provide a further discussion and extension of this work. In Section B, we provide related works. In Section C, we provide general definitions and several basic facts. In Section D, we show how we calculate the gradient of the loss function. In Section E, we show the time complexity of our algorithm. In Section F, we show that our algorithm can be computed in polynomial time. In Section G, we show the hardness of our algorithm. In Section **??**, we discuss the limitation of this work. In Section **??**, we provide an elaborate discussion about potential societal impacts.

## A    FURTHER DISCUSSION AND EXTENSION

**Technical novelty over previous works.** We generalize beyond the results of (Alman & Song, 2024b), which only provide methods for *tensor attention forward*. Our paper presents a detailed analysis for *tensor attention backward*, providing both upper bound and lower bound. Though we build on some results from (Alman & Song, 2024b) and (Alman & Song, 2024a), generalizing to tensor attention backward posed many technical challenges. These challenges are unique to our setting and not presented in previous settings like matrix attention (Alman & Song, 2023; 2024a) or tensor attention forward (Alman & Song, 2024b). To be more specific, we prove many key properties for the tensor operation needed for backward though not needed for forward, including 4.4 (distribution rule for tensor and matrix product), C.11 (tensor computation reduction to matrix product), C.12 (distribution rule for tensor computation), Claim C.20 (tensor product to matrix product). Lemma 4.5 supports the proof of Fact 4.4 and helps bypass the $O(n^3 d^2)$ time complexity bottleneck in the fast computation of $U_2$. Fact 4.4, crucial in proving Lemma F.5, shows the distributive nature of tensor operations. Using Facts C.11, C.12, and Claim C.20, we leverage the structure of low-rank matrices $U_5, V_5, W_5$ to prove Theorem 4.2.

**Connection to real applications.** There are some empirical studies attempting to implement similar tensor attention (three order) in language modeling (Ma et al., 2019) and 3D medical image segmentation (Wang et al., 2023). However, due to cubic time complexity, their models remain relatively small, e.g, 12M parameters in (Ma et al., 2019). Although small scale, (Ma et al., 2019; Wang et al., 2023) demonstrates the significant potential of tensor attention. Our work proves that an almost linear time algorithm for tensor attention mechanisms exists (Algorithm 1). This advancement could enable the scaling up of tensor attention and facilitate novel model designs in multi-modality, 3D imaging, and beyond. On the other hand, we abstract the most challenging part (the highest time complexity operation) in high-order attention into a clear mathematical problem and provide a solution. Our work introduces a new concept to the community, suggesting that cubic time complexity may not be the bottleneck in implementing three-order attention during training. Practical implementation poses additional significant challenges, considering numerous other techniques and operations, such as dropout, layer normalization, residual connections, position encoding, and many others. We hope our work inspires further algorithmic design.

**Feasibility when the large value exists in the matrices.** If there exist many large entries in $Q, K_1, K_2, V_1, V_2$, our hardness results (Theorem 5.3) indicate that no algorithm can accelerate the attention computation. However, several exciting works (Sun et al., 2024; Han et al., 2024) have shown that large entries are very sparse in the attention matrix. This suggests that our Algorithm 1 could inspire many potential practical implementations. One straightforward approach is to handle large entries separately, as in (Han et al., 2024), and apply our algorithm to the remaining parts. There is undoubtedly a broad algorithm design space, and we hope our work provides valuable insights.

**Extend our technique to compute the module-wise gradient.** Let $n$ be the input toke length, and $d$ be the hidden dimension. At the $i$-th layer of transformer model, let $G_i \in \mathbb{R}^{n \times d}$ denote the output of upstream gradient, $X_i \in \mathbb{R}^{n \times d}$ be defined in Definition 2.8, and $\mathsf{Attn}_i := D^{-1} A V$ be the tensor attention model where $D, A, V$ are defined in Definition 2.5. Let $\mathsf{Loss}$ be some loss function. Then, by the chain rule, we have the module-wise gradient $\frac{\mathrm{d} \mathsf{Loss}}{\mathrm{d} X_i} = \mathrm{vec}(G_i) \frac{\mathrm{d} \mathsf{Attn}_i}{\mathrm{d} X_i}$.

**Extend our technique to the multi-head attention.** The gradient computation for each attention head in the same layer is independent of the others; each head only depends on its upstream gradient and its current module-wise gradient according to the chain rule. Therefore, our analysis can be directly applied to multi-head attention.

**Generalize to scenarios involving multiple modalities** In our three-order attention, one attention module can handle three modalities simultaneously, i.e., $Q, K_1, K_2$. For more modality, e.g., $m > 3$ modality, there are two potential solutions in our minds. First, we could use $m$-order attention, i.e., $Q, K_1, K_2, \ldots, K_{m-1}$. The inference and training time complexity for this approach are still unknown, and we leave it as our future work. Second, we could use multiple modules of three-order attention. Note that one layer of standard attention may introduce one more modality $K_1$ each time, while one layer of three-order attention may introduce two more modalities $K_1, K_2$ each time. Thus, if we have $m + 1$ modality and $Q$ is from one modality, say text, then the standard attention may need $m$ layers to merge all modalities together, whereas three-order attention may only need $\log(m)$ layers to merge them all together.

## B  RELATED WORK

**Fast attention computation.** In recent years, significant advances have been made in the development of efficient attention computation. One research direction involves employing low-rank approximations, polynomial kernel, or random features for the attention matrix (Choromanski et al., 2020; Zheng et al., 2022; Alman & Song, 2023; Kacham et al., 2023; Song et al., 2024; Gu et al., 2024), which scales the computational complexity sub-quadratically with sequence length. Another method explores patterns of sparse attention that lessen the computational load (Han et al., 2024). Additionally, using linear attention as an alternative to softmax attention has emerged as a substantial area of study (Katharopoulos et al., 2020; Schlag et al., 2021; Zhang et al., 2023b; Ahn et al., 2024; Zhang et al., 2024). These innovations have enhanced the capability of transformer-based models to handle longer sequences, thereby broadening their potential applications across various fields (Chen et al., 2023; Su et al., 2024; Peng et al., 2024; Ding et al., 2024; Ma et al., 2024; Bertsch et al., 2023; Jin et al., 2024). On the other hand, FlashAttention (Dao et al., 2022; Dao, 2024) is one of the most popular practical methods to accelerate attention computation, while it achieves a considerable improvement in running time with a constant complexity ratio.

**Tensor computation for high-order representation.** Tensors excel over matrices in capturing higher-order relationships within data (Zhang et al., 2025). Calculating low-rank factorizations or approximations of tensors is essential in a wide range of computer science applications, such as natural language processing (Lei et al., 2015; Bouchard et al., 2015), computer vision (Lu et al., 2016; Chen et al., 2017), computer graphics (Wang et al., 2005; Vasilescu, 2009), security (Acar et al., 2006; Kolda & Bader, 2006), and data mining (Karatzoglou et al., 2010; Rendle & Schmidt-Thieme, 2010; Mørup, 2011). Moreover, tensors are crucial in numerous machine learning applications (Podosinnikova et al., 2015; Zhong et al., 2017; Yang et al., 2019) and other diverse fields (Reps et al., 2016; Yi et al., 2016; Ray et al., 2016).

**Large language models and transformer.** The foundation of the success of generative large language models (LLMs) lies in the decoder-only transformer architecture, as introduced by (Vaswani et al., 2017). This architecture has become critical for many leading models in natural language processing (NLP) (Chang et al., 2024). These models have already demonstrated their capabilities in various real-world applications, including language translation (He et al., 2021), sentiment analysis (Usama et al., 2020), and language modeling (Martin et al., 2019), due to their emergent ability, e.g., compositional ability (Dziri et al., 2024; Xu et al., 2024; Li et al., 2024), in-context learning (Olsson et al., 2022; Min et al., 2022; Shi et al., 2023). The transformer leverages a self-attention mechanism, which enables the model to identify long-range dependencies within the input sequence. Self-attention calculates a weighted sum of input tokens, with weights based on the similarity between token pairs. This allows the model to focus on pertinent information from various parts of the sequence during output generation.

## C    Tensor Operation Background

In Section C.1, we define the notation of computational time and the tensor operation. In Section C.2, we provide some helpful facts of tensor operation. In Section C.3, we provide some helpful facts of vectorization operation. In Section C.4, we provide some helpful facts about the tensor product. It is worth noting that proofs for some of the facts discussed in this section are also available in (Kolda & Bader, 2009).

### C.1    General definitions and tensor operation

**Fact C.1** ((Bürgisser et al., 2013; Bläser, 2013)). *We can show that* $\mathcal{T}_{\mathrm{mat}}(a, b, c) = O(\mathcal{T}_{\mathrm{mat}}(a, c, b)) = O(\mathcal{T}_{\mathrm{mat}}(b, a, c)) = O(\mathcal{T}_{\mathrm{mat}}(b, c, a)) = O(\mathcal{T}_{\mathrm{mat}}(c, a, b)) = O(\mathcal{T}_{\mathrm{mat}}(c, b, a))$.

We define the third mode tensor product, which is the core operator of tensor operations.

**Definition C.2** (Third mode tensor product $(\cdot, \cdot, \cdot)$). *Let* $\mathsf{X} \in \mathbb{R}^{d \times d \times d}$. *Given matrices* $A_1 \in \mathbb{R}^{n \times d}$, $A_2 \in \mathbb{R}^{n \times d}$ *and* $A_3 \in \mathbb{R}^{n \times d}$. *Let operator* $\mathsf{X}(A_1, A_2, A_3) \in \mathbb{R}^{n \times n \times n}$ *satisfying*

$$\mathsf{X}(A_1, A_2, A_3)_{i,j,l} := \sum_{a=1}^{d} \sum_{b=1}^{d} \sum_{c=1}^{d} \mathsf{X}_{a,b,c}(A_1)_{i,a}(A_2)_{j,b}(A_3)_{l,c}, \quad \forall i \in [n], j \in [n], l \in [n].$$

**Definition C.3** ($\odot$ tensor computation). *Given matrices* $A \in \mathbb{R}^{n \times d}$, $B \in \mathbb{R}^{n \times d}$, $C \in \mathbb{R}^{n \times d}$, *we use* $T = A \odot B \odot C \in \mathbb{R}^{n \times n \times n}$ *to denote an tensor whose entries are given by*

$$T_{i,j,l} := \sum_{a=1}^{d} A_{i,a} B_{j,a} C_{l,a}, \quad \forall i \in [n], j \in [n], l \in [n].$$

We note that a tensor $T$ can be written in the form $A \odot B \odot C$ like this if and only if its tensor rank is at most $d$.

### C.2    Facts for tensor operation

**Fact C.4** (Transpose rule). *We show the results below*

- *Suppose that* $\underbrace{K}_{n_1 n_2 \times d} = \underbrace{K_1}_{n_1 \times d} \oslash \underbrace{K_2}_{n_2 \times d}$. *We have* $\underbrace{K^\top}_{d \times n_1 n_2} = \underbrace{K_1^\top}_{d \times n_1} \ominus \underbrace{K_2^\top}_{d \times n_2}$.

- *Suppose that* $\underbrace{Q}_{n \times d_1 d_2} = \underbrace{Q_1}_{n \times d_1} \ominus \underbrace{Q_2}_{n \times d_2}$. *We have* $\underbrace{Q^\top}_{d_1 d_2 \times n} = \underbrace{Q_1^\top}_{d_1 \times n} \oslash \underbrace{Q_2^\top}_{d_2 \times n}$.

- *Suppose that* $\underbrace{V}_{n_1 n_2 \times d_1 d_2} = \underbrace{V_1}_{n_1 \times d_1} \otimes \underbrace{V_2}_{n_2 \times d_2}$. *We have* $\underbrace{V^\top}_{d_1 d_2 \times n_1 n_2} = \underbrace{V_1^\top}_{d_1 \times n_1} \otimes \underbrace{V_2^\top}_{d_2 \times n_2}$.

*Proof.* The proof is very straightforward. $\qquad\square$

**Fact C.5** (Swap rule). *Let* $V_1 \in \mathbb{R}^{n \times d}$. *Let* $V_2 \in \mathbb{R}^{n \times k}$. *Let* $W_1 \in \mathbb{R}^{m \times d}$. *Let* $W_2 \in \mathbb{R}^{m \times k}$. *We can show swap rule for* $\oslash$ *and* $\ominus$,

$$\underbrace{(V_1 \ominus V_2)}_{n \times dk} \oslash \underbrace{(W_1 \ominus W_2)}_{m \times dk} = \underbrace{(V_1 \oslash W_1)}_{mn \times d} \ominus \underbrace{(V_2 \oslash W_2)}_{mn \times k}$$

*And we can show swap rule for* $\otimes$ *and* $\ominus$,

$$\underbrace{(V_1 \ominus V_2)}_{n \times dk} \otimes \underbrace{(W_1 \ominus W_2)}_{m \times dk} = \underbrace{(V_1 \otimes W_1)}_{mn \times dk} \ominus \underbrace{(V_2 \otimes W_2)}_{mn \times dk}$$

*Proof.* The proof is trivially following from definition of $\oslash$ and $\ominus$.

Note that for any $i_1 \in [n], i_2 \in [m], j_1 \in [d], j_2 \in [k]$

$$((V_1 \ominus V_2) \oslash (W_1 \ominus W_2))_{i_1+(i_2-1)n, j_1+(j_2-1)d}$$
$$= (V_1)_{i_1,j_1}(V_2)_{i_1,j_2}(W_1)_{i_2,j_1}(W_2)_{i_2,j_2}$$
$$= ((V_1 \oslash W_1) \ominus (V_2 \oslash W_2))_{i_1+(i_2-1)n, j_1+(j_2-1)d}$$

Thus, we complete the proof. $\square$

**Remark C.6.** *In Fact C.5, due to definition $V_1$ and $V_2$ need to have the same number of rows. $W_1$ and $W_2$ also need to have the same number of rows. $V_1$ and $W_1$ need to have same number of columns, and $V_2$ and $W_2$ need to have same number of columns.*

**Fact C.7** (Swap rule for tensor product and matrix product). *Let $W_1, W_2 \in \mathbb{R}^{d \times d}$ and $A_1, A_2 \in \mathbb{R}^{n \times d}$. We have*

$$\underbrace{(A_1 \otimes A_2)}_{n^2 \times d^2} \cdot \underbrace{(W_1 \oslash W_2)}_{d^2 \times d} = \underbrace{(A_1 \cdot W_1)}_{n \times d} \oslash \underbrace{(A_2 \cdot W_2)}_{n \times d}.$$

*Proof.* For any $i_1, i_2 \in [n], j \in [d]$, we have

$$((A_1 \otimes A_2) \cdot (W_1 \oslash W_2))_{i_1+(i_2-1)n, j}$$
$$= \sum_{k_1 \in [d], k_2 \in [d]} (A_1 \otimes A_2)_{i_1+(i_2-1)n, k_1+(k_2-1)d}(W_1 \oslash W_2)_{k_1+(k_2-1)d, j}$$
$$= \sum_{k_1 \in [d], k_2 \in [d]} (A_1 \otimes A_2)_{i_1+(i_2-1)n, k_1+(k_2-1)d} \cdot (W_1)_{k_1,j} \cdot (W_2)_{k_2,j}$$
$$= \sum_{k_1 \in [d], k_2 \in [d]} (A_1)_{i_1,k_1} \cdot (A_2)_{i_2,k_2} \cdot (W_1)_{k_1,j} \cdot (W_2)_{k_2,j}$$
$$= (\sum_{k_1 \in [d]} (A_1)_{i_1,k_1} \cdot (W_1)_{k_1,j}) \cdot (\sum_{k_2 \in [d]} (A_2)_{i_2,k_2} \cdot (W_2)_{k_2,j})$$
$$= (A_1 \cdot W_1)_{i_1,j} \cdot (A_2 \cdot W_2)_{i_2,j}$$
$$= ((A_1 \cdot W_1) \oslash (A_2 \cdot W_2))_{i_1+(i_2-1)n, j},$$

where the first step follows matrix multiplication, the second step follows Definition 2.2, the third step follows Definition 2.1, the fourth step follows simple algebra, the fifth step follows matrix multiplication and the last step follows Definition 2.2. $\square$

**Fact C.8** (Restatement of Fact 4.4). *Let $U_1 \in \mathbb{R}^{n_1 \times d}$ and $U_2 \in \mathbb{R}^{n_1 \times k}$. Let $V_1 \in \mathbb{R}^{n_2 \times d}$ and $V_2 \in \mathbb{R}^{n_2 \times k}$. Let $W_1 \in \mathbb{R}^{n_3 \times d}$ and $W_2 \in \mathbb{R}^{n_3 \times k}$. We have*

$$\underbrace{(U_1 \ominus U_2)}_{n_1 \times dk} \cdot (\underbrace{(V_1 \ominus V_2)}_{n_2 \times dk} \oslash \underbrace{(W_1 \ominus W_2)}_{n_3 \times dk})^\top = (\underbrace{U_1}_{n_1 \times d} (\underbrace{V_1}_{n_2 \times d} \oslash \underbrace{W_1}_{n_3 \times d})^\top) \circ (\underbrace{U_2}_{n_1 \times k} (\underbrace{V_2}_{n_2 \times k} \oslash \underbrace{W_2}_{n_3 \times k})^\top)$$

*Proof of Fact 4.4.* We can show that

$$(U_1 \ominus U_2)((V_1 \ominus V_2) \oslash (W_1 \ominus W_2))^\top = (U_1 \ominus U_2)((V_1 \oslash W_1) \ominus (V_2 \oslash W_2))^\top$$
$$= (U_1 \ominus U_2)((V_1 \oslash W_1)^\top \oslash (V_2 \oslash W_2)^\top)$$
$$= (U_1^\top \oslash U_2^\top)^\top((V_1 \oslash W_1)^\top \oslash (V_2 \oslash W_2)^\top)$$
$$= (U_1(V_1 \oslash W_1)^\top) \circ (U_2(V_2 \oslash W_2)^\top)$$

where first step is due to swapping rule for $\oslash$ and $\ominus$ (see Fact C.5), the second step follows from Fact C.4, the third step follows from Fact C.4, and the last step follows from Lemma C.13. $\square$

**Fact C.9.** *Let $U_1 \in \mathbb{R}^{n_1 \times d^2}$ and $U_2 \in \mathbb{R}^{n_1 \times k^2}$. Let $V_1 \in \mathbb{R}^{n_2 \times d}$ and $V_2 \in \mathbb{R}^{n_2 \times k}$. Let $W_1 \in \mathbb{R}^{n_3 \times d}$ and $W_2 \in \mathbb{R}^{n_3 \times k}$. We have*

$$\underbrace{(U_1 \ominus U_2)}_{n_1 \times d^2 k^2} \cdot (\underbrace{(V_1 \ominus V_2)}_{n_2 \times dk} \otimes \underbrace{(W_1 \ominus W_2)}_{n_3 \times dk})^\top = (\underbrace{U_1}_{n_1 \times d^2} (\underbrace{V_1}_{n_2 \times d} \otimes \underbrace{W_1}_{n_3 \times d})^\top) \circ (\underbrace{U_2}_{n_1 \times k^2} (\underbrace{V_2}_{n_2 \times k} \otimes \underbrace{W_2}_{n_3 \times k})^\top)$$

*Proof.* We can show that,

$$\underbrace{(U_1 \ominus U_2)}_{n_1 \times d^2 k^2} \cdot (\underbrace{(V_1 \ominus V_2)}_{n_2 \times dk} \otimes \underbrace{(W_1 \ominus W_2)}_{n_3 \times dk}))^\top$$

$$= \underbrace{(U_1 \ominus U_2)}_{n_1 \times d^2 k^2} \cdot ((V_1 \otimes W_1) \ominus (V_2 \otimes W_2))^\top$$

$$= (U_1 \ominus U_2) \cdot ((V_1 \otimes W_1)^\top \oslash (V_2 \otimes W_2)^\top)$$

$$= (U_1^\top \oslash U_2^\top)^\top \cdot ((V_1 \otimes W_1)^\top \oslash (V_2 \otimes W_2)^\top)$$

$$= (\underbrace{U_1}_{n_1 \times d^2} (\underbrace{V_1}_{n_2 \times d} \otimes \underbrace{W_1}_{n_3 \times d})^\top) \circ (\underbrace{U_2}_{n_1 \times k^2} (\underbrace{V_2}_{n_2 \times k} \otimes \underbrace{W_2}_{n_3 \times k})^\top)$$

where the first step is because of the swap rule for $\otimes$ and $\ominus$ (see Fact C.5), the second step follows from Fact C.4, the third step follows from Fact C.4, and the last step follows from Lemma C.13. $\quad\square$

**Claim C.10.** *Let $A, B, C \in \mathbb{R}^{n \times d}$.*

**Part 1.** *Let $I_d \in \mathbb{R}^{d \times d}$ denote an identity matrix. Then, we have*

$$A I_d B^\top = A B^\top.$$

**Part 2.** *Let $\mathsf{I}_d \in \mathbb{R}^{d \times d \times d}$ denote an identity tensor. Then we can show that*

$$\mathsf{I}_d(A, B, C) = A \odot B \odot C$$

*Proof.* Now we prove for each part.

**Proof of Part1.** Using the property of identity matrix, it's easy to see this holds.

**Proof of Part2.**

$$\mathsf{I}_d(A, B, C) = \sum_{a=1}^d \sum_{b=1}^d \sum_{c=1}^d (\mathsf{I}_d)_{a,b,c} (A)_{i,a} (B)_{j,b} (C)_{l,c}$$

$$= \sum_{a=1}^d (A)_{i,a} (B)_{j,a} (C)_{l,a}$$

$$= A \odot B \odot C$$

where the first step follows from Definition C.2, the second step follows from the property of identity tensor $(\mathsf{I}_d)_{i,j,k}$, which equals 1 only when $i = j = k$ and 0 elsewhere, and the last step follows from Definition C.3. $\quad\square$

**Fact C.11.** *Let $U, V, W \in \mathbb{R}^{n \times d}$, we have*

$$\underbrace{U(V \oslash W)^\top}_{n \times n^2} = \underbrace{\mathsf{mat}(U \odot V \odot W)}_{n \times n^2}.$$

*Proof.* For any $i, j, k \in [n]$, we have

$$\mathsf{mat}(U \odot V \odot W)_{i,(j-1)n+k} = (U \odot V \odot W)_{i,j,k}$$

$$= \sum_{a \in [d]} U_{i,a} V_{j,a} W_{k,a}$$

$$= \sum_{a \in [d]} U_{i,a} (V \oslash W)_{(j-1)n+k,a}$$

$$= \sum_{a \in [d]} U_{i,a} ((V \oslash W)^\top)_{a,(j-1)n+k}$$

$$= (U(V \oslash W)^\top)_{i,(j-1)n+k},$$

where the first step by definition of mat, the second step follows Definition C.3, the third step follows Definition 2.2, the fourth step follows from transpose, and the last step follows from matrix multiplication. $\quad\square$

**Fact C.12.** *Given $A_1, A_2, A_3 \in \mathbb{R}^{n \times d}$ and $W_1, W_2, W_3 \in \mathbb{R}^{n \times k}$, we have*

$$\underbrace{[W_1 \odot W_2 \odot W_3](A_1^\top, A_2^\top, A_3^\top)}_{d \times d \times d} = \underbrace{((A_1^\top W_1) \odot (A_2^\top W_2) \odot (A_3^\top W_3))}_{d \times d \times d}.$$

*Proof.* The proof is trivial by Definition C.3 and Definition C.2. $\qquad\square$

We prove an important tool, which will be used in analyzing the running time of our algorithm.

**Lemma C.13** ( Formal version of Lemma 4.5 )**.** *If the following condition holds*

- *Let $\oslash$ be defined as Definition 2.2.*

- *Given $A_1 \in \mathbb{R}^{n_1 \times d_1}$, $A_2 \in \mathbb{R}^{n_2 \times d_1}$, let $A := (A_1 \oslash A_2) \in \mathbb{R}^{n_1 n_2 \times d_1}$.*

- *Given $B_1 \in \mathbb{R}^{n_1 \times d_2}$, $B_2 \in \mathbb{R}^{n_2 \times d_2}$, let $B := (B_1 \oslash B_2) \in \mathbb{R}^{n_1 n_2 \times d_2}$.*

- *We define $C \in \mathbb{R}^{d_1 \times d_2}$ as $C := A^\top B$*

- *We define $\underbrace{C_1}_{d_1 \times d_2} := A_1^\top B_1$, $\underbrace{C_2}_{d_1 \times d_2} := A_2^\top B_2$*

*Then, we have*

- *Part 1. $C_1 \circ C_2 = C$*

- *Part 2. Given as input $A_1, A_2, B_1, B_2$, we can get $C$ in $\mathcal{T}_{\mathrm{mat}}(d_1, \max\{n_1, n_2\}, d_2)$ time.*

*Proof.* For each $i \in [n_1]$, let $a_{1,i}^\top$ denote the $i$-th row of $A_1 \in \mathbb{R}^{n_1 \times d_1}$.

For each $i \in [n_2]$, let $a_{2,i}^\top$ denote the $i$-th row of $A_2 \in \mathbb{R}^{n_2 \times d_1}$.

For each $i \in [n_1]$, let $b_{1,i}^\top$ denote the $i$-th row of $B_1 \in \mathbb{R}^{n_1 \times d_2}$.

For each $i \in [n_2]$, let $b_{2,i}^\top$ denote the $i$-th row of $B_2 \in \mathbb{R}^{n_2 \times d_2}$.

Recall that $C_1 \in \mathbb{R}^{d_1 \times d_2}$ and $C_2 \in \mathbb{R}^{d_1 \times d_2}$,

$$C_1 := A_1^\top B_1, \quad C_2 := A_2^\top B_2$$

Thus, we see that for all $\forall k_1 \in [d_1], k_2 \in [d_2]$

$$(C_1)_{k_1, k_2} = \sum_{i=1}^{n_1} a_{1,i,k_1} b_{1,i,k_2}$$

$$(C_2)_{k_1, k_2} = \sum_{j=1}^{n_2} a_{2,j,k_1} b_{2,j,k_2}$$

Then, we can write $C \in \mathbb{R}^{d_1 \times d_2}$ as

$$\underbrace{C}_{d_1 \times d_2} = \underbrace{A^\top}_{d_1 \times n_1 n_2} \underbrace{B}_{n_1 n_2 \times d_2}$$

$$= \sum_{i=1}^{n_1 n_2} \underbrace{A_{i,*}}_{d_1 \times 1} \underbrace{(B_{i,*})^\top}_{1 \times d_2}$$

$$= \sum_{i=1}^{n_1} \sum_{j=1}^{n_2} \underbrace{A_{i+(j-1)n_1, *}}_{d_1 \times 1} \cdot \underbrace{(B_{i+(j-1)n_1, *})^\top}_{1 \times d_2}$$

$$= \sum_{i=1}^{n_1} \sum_{j=1}^{n_2} \underbrace{(a_{1,i} \circ a_{2,j})}_{d_1 \times 1} \cdot \underbrace{(b_{1,i} \circ b_{2,j})^\top}_{1 \times d_2} \qquad (1)$$

where the first step follows from definition of $C \in \mathbb{R}^{d \times d}$, the second step follows from the matrix can written as the summation of $n_1 n_2$ rank-1 matrices, the third step follows from changing the index, the forth step follows from $\underbrace{A_{i+(j-1)n_1,*}}_{d_1 \times 1} = \underbrace{a_{1,i}}_{d_1 \times 1} \circ \underbrace{a_{2,j}}_{d_1 \times 1}$ by Definition 2.2.

From the above, we can calculate that the entry of $C$ in location $k_1 \in [d_1], k_2 \in [d_2]$ is

$$C_{k_1,k_2} = \sum_{i=1}^{n_1} \sum_{j=1}^{n_2} (a_{1,i} \circ a_{2,j})_{k_1} \cdot (b_{1,i} \circ b_{2,j})_{k_2}^{\top}$$

$$= \sum_{i=1}^{n_1} \sum_{j=1}^{n_2} a_{1,i,k_1} a_{2,j,k_1} b_{1,i,k_2} b_{2,j,k_2}$$

$$= (\sum_{i=1}^{n_1} a_{1,i,k_1} b_{1,i,k_2}) \cdot (\sum_{j=1}^{n_2} a_{2,j,k_1} b_{2,j,k_2})$$

$$= (C_1)_{k_1,k_2} \cdot (C_2)_{k_1,k_2}$$

where the first step follows from Eq. (1), the second step follows from simple algebra, the third step follows from separating the summation over $i$ and the summation over $j$, and the last step follows from definition of matrices $C_1$ and $C_2$.

Thus, we can conclude

$$C = C_1 \circ C_2.$$

The algorithm will first compute $C_1$ and $C_2$, which takes $\mathcal{T}_{\mathrm{mat}}(d_1, \max\{n_1, n_2\}, d_2)$ time. Then it calculates $C_1 \circ C_2$, which takes $O(d_1 d_2)$ time. $\qquad\square$

### C.3  FACTS FOR VECTORIZATION OPERATION

**Fact C.14.** *Let $A, B \in \mathbb{R}^{n \times d}$. Then,*

$$\mathrm{tr}[A^{\top} B] = \mathrm{vec}(A)^{\top} \mathrm{vec}(B)$$

*Proof.* We can show

$$\mathrm{tr}[A^{\top} B] = \sum_{i=1}^{n} \sum_{j=1}^{d} A_{i,j} B_{i,j}$$

$$= \mathrm{vec}(A)^{\top} \mathrm{vec}(B)$$

where the first step is due to the definition of trace, and the second step is because of the definition of vec operator. $\qquad\square$

**Fact C.15.** *Let $a \in \mathbb{R}^n, b \in \mathbb{R}^d$. Then,*

$$\mathrm{vec}(ab^{\top}) = a \otimes b$$

*Proof.* We can show

$$\mathrm{vec}(ab^{\top}) = \mathrm{vec}(\begin{bmatrix} a_1 b^{\top} \\ a_2 b^{\top} \\ \dots \\ a_n b^{\top} \end{bmatrix})$$

$$= [a_1 b^{\top}, a_2 b^{\top}, \dots, a_n b^{\top}]^{\top}$$

$$= a \otimes b$$

where the first step follows from the definition of the outer product, the second step follows from the definition of vectorization operator $\mathrm{vec}(\cdot)$ which stacks rows of a matrix into a column vector, and the last step follows from Definition 2.1. $\qquad\square$

**Fact C.16** (Tensor-trick, Restatement of Fact 4.6). *Given matrices $A_1 \in \mathbb{R}^{n_1 \times d_1}$, $A_2 \in \mathbb{R}^{n_2 \times d_2}$ and $X \in \mathbb{R}^{d_1 \times d_2}$, we have $\mathrm{vec}(A_1 X A_2^\top) = (A_1 \otimes A_2) \mathrm{vec}(X) \in \mathbb{R}^{n_1 n_2}$.*

*Proof of Fact 4.6.* We can show

$$\mathrm{vec}(A_1 X A_2^\top) = \sum_{i=1}^{d_1} \sum_{j=1}^{d_2} X_{i,j} \mathrm{vec}(A_{1,*,i}(A_{2,*,j})^\top)$$

$$= \sum_{i=1}^{d_1} \sum_{j=1}^{d_2} X_{i,j} (\underbrace{A_{1,*,i}}_{n_1 \times 1} \otimes \underbrace{A_{2,*,j}}_{n_2 \times 1})$$

$$= \sum_{i=1}^{d_1} (\underbrace{A_{1,*,i}}_{n_1 \times 1} \otimes \underbrace{A_2}_{n_2 \times d_2}) \underbrace{X_{i,*}}_{d_2 \times 1}$$

$$= (A_1 \otimes A_2) \mathrm{vec}(X)$$

where the first step is due to the matrix being able to be written as a summation of vectors, the second step follows from Fact C.15, the third step follows from that matrix can be written as a summation of vectors, and the last step follows from the definition of vectorization operator $\mathrm{vec}(\cdot)$. $\square$

**Fact C.17.** *Let $A \in \mathbb{R}^{n_1 \times n_2}$, $B \in \mathbb{R}^{n_2 \times n_3}$, $C \in \mathbb{R}^{n_3 \times n_4}$, $D \in \mathbb{R}^{n_4 \times n_5}$.*

*We have*

$$\mathrm{tr}[ABCD] = \mathrm{vec}(A^\top)^\top (B \otimes D^\top) \mathrm{vec}(C)$$

*Proof.* We can show

$$\mathrm{tr}[ABCD] = \mathrm{vec}(A^\top)^\top \mathrm{vec}(BCD)$$
$$= \mathrm{vec}(A^\top)^\top (B \otimes D^\top) \mathrm{vec}(C)$$

where the first step follows from Fact C.14, and the second step follows from Fact C.16. $\square$

**Fact C.18.** *Let $A, B \in \mathbb{R}^{n \times n}$ be two $n \times n$ symmetric matrices. Let $X$ and $Y$ denote two $n \times n$ matrices. Then we have*

$$\mathrm{vec}(A)^\top (X \otimes Y) \mathrm{vec}(B) = \mathrm{vec}(A)^\top (Y \otimes X) \mathrm{vec}(B)$$

*Proof.* We can show that

$$\mathrm{vec}(A)^\top (X \otimes Y) \mathrm{vec}(B) = \mathrm{tr}[A^\top X B Y^\top]$$
$$= \mathrm{tr}[B Y^\top A^\top X]$$
$$= \mathrm{vec}(B^\top)^\top (Y^\top \otimes X^\top) \mathrm{vec}(A^\top)$$
$$= \mathrm{vec}(B)^\top (Y^\top \otimes X^\top) \mathrm{vec}(A)$$
$$= ((Y^\top \otimes X^\top) \mathrm{vec}(A))^\top \mathrm{vec}(B)$$
$$= \mathrm{vec}(A)^\top (Y \otimes X) \mathrm{vec}(B)$$

where the first step follows from Fact C.17, the second step follows from the cyclic property of trace, the third step follows from Fact C.17, the fourth step follows from $A, B$ is symmetric, the fifth step is due to the definition of inner product, and the last step is due to Fact C.4. $\square$

## C.4 FACTS FOR TENSOR PRODUCT

**Fact C.19.** *Let $X = \underbrace{\mathrm{mat}(\mathsf{I}_d)}_{d \times d^2}$, where $\mathsf{I}_d \in \mathbb{R}^{d \times d \times d}$ and $A_1, A_2 \in \mathbb{R}^{n \times d}$. We have*

$$\underbrace{(A_1 \otimes A_2)}_{n^2 \times d^2} \underbrace{X^\top}_{d^2 \times d} = \underbrace{A_1 \oslash A_2}_{n^2 \times d}.$$

*Proof.* For any $i_1, i_2 \in [n], j \in [d]$, we have

$$((A_1 \otimes A_2)X^\top)_{i_1+(i_2-1)n, j} = \sum_{k_1 \in [d], k_2 \in [d]} (A_1 \otimes A_2)_{i_1+(i_2-1)n, k_1+(k_2-1)d} X_{j, k_1+(k_2-1)d}$$

$$= \sum_{k_1 \in [d], k_2 \in [d]} (A_1)_{i_1, k_1} \cdot (A_2)_{i_2, k_2} X_{j, k_1+(k_2-1)d}$$

$$= (A_1)_{i_1, j} \cdot (A_2)_{i_2, j}$$

$$= (A_1 \oslash A_2)_{i_1+(i_2-1)n, j},$$

where the first step is due to matrix multiplication, the second step follows Definition 2.1, the third step follows $X_{j, k_1+(k_2-1)d} = 1$ when $j = k_1 = k_2$, and $X_{j, k_1+(k_2-1)d} = 0$ otherwise, and the last step is because of Definition 2.2. $\square$

**Claim C.20.** *Given* $X \in \mathbb{R}^{d \times d^2}$. *Note* $\mathsf{X} \in \mathbb{R}^{d \times d \times d}$ *denotes its tensor version. Given matrices* $A_1, A_2, A_3 \in \mathbb{R}^{n \times d}$. *Following Definition C.2, we can show*

$$\big(\underbrace{A_1}_{n \times d} \underbrace{X}_{d \times d^2} \underbrace{(A_2 \otimes A_3)^\top}_{d^2 \times n^2}\big)_{i, (j-1)n+l} = (\underbrace{\mathsf{X}(A_1, A_2, A_3)}_{n \times n \times n})_{i, j, l}, \quad \forall i \in [n], j \in [n], l \in [n]$$

*and*

$$\mathrm{vec}(\underbrace{A_1}_{n \times d} \underbrace{X}_{d \times d^2} \underbrace{(A_2 \otimes A_3)^\top}_{d^2 \times n^2}) = \mathrm{vec}(\underbrace{\mathsf{X}(A_1, A_2, A_3)}_{n \times n \times n}).$$

*Proof.* We can show that

$$(A_1 X (A_2 \otimes A_3)^\top)_{i, (j-1)n+l} = \sum_{a=1}^d \sum_{b=1}^d \sum_{c=1}^d (A_1)_{i,a} X_{a, (b-1)d+c} (A_2)_{j,b} (A_3)_{l,c}$$

$$= \sum_{a=1}^d \sum_{b=1}^d \sum_{c=1}^d \mathsf{X}_{a,b,c} (A_1)_{i,a} (A_2)_{j,b} (A_3)_{l,c}$$

$$= \mathsf{X}(A_1, A_2, A_3)_{i,j,l},$$

where the first step follows the Kronecker product Definition 2.1, the second step follows $\mathsf{X}_{a,b,c} = X_{a, (b-1)d+c}$, and the last step is due to Definition C.2. $\square$

Now, we introduce a key claim that can reduce the tensor product to matrix multiplication and Kronecker product to make calculation easy.

**Claim C.21.** *Let* $\mathsf{I}_d \in \mathbb{R}^{d \times d \times d}$ *and* $A_1, A_2, A_3 \in \mathbb{R}^{n \times d}$. *We have* $\mathrm{mat}(\mathsf{I}_d(A_1, A_2, A_3)) = A_1 \mathrm{mat}(\mathsf{I}_d)(A_2 \otimes A_3)^\top = A_1(A_2 \oslash A_3)^\top \in \mathbb{R}^{n \times n^2}$.

*Proof.* The proof follows from Claim C.20 and Fact C.19. $\square$

# D  GRADIENT FORMULATION AND ANALYSIS

In Section D.1, we define some useful function that will help further calculation. In Section D.2, we define the expression for the loss function. In Section D.3, we give detailed gradient computation.

## D.1  DEFINITIONS FOR USEFUL FUNCTIONS

We will introduce the definition of $\mathsf{K}$, $\alpha$, $\mathsf{S}$, and $\mathsf{L}$ used in loss formulation.

**Definition D.1.** *We define* $A_1, A_2, A_3 \in \mathbb{R}^{n \times d}$ *to be three matrices in size* $n \times d$. *Suppose that* $\mathsf{A} = A_1 \otimes A_2 \otimes A_3 \in \mathbb{R}^{n^3 \times d^3}$. *Let* $\mathsf{A}_{j_0} \in \mathbb{R}^{n^2 \times d^3}$ *represent an* $n^2 \times d^3$ *sub-block from* $\mathsf{A}$. *There are* $n$ *such sub-blocks, i.e. the* $(i + (j_0 - 1) \cdot n^2)$-*th row, $j$-th column of* $\mathsf{A}$ *is the $i$-th row, $j$-th column of* $\mathsf{A}_{j_0}$, *for* $i \in [n^2], j \in [d^3], j_0 \in [n]$.

*For all $j_0 \in [n]$, we denote function $\mathsf{K}(x)_{j_0} : \mathbb{R}^{d^3} \to \mathbb{R}^{n^2}$ as below:*

$$\mathsf{K}(x)_{j_0} := \underbrace{\exp(\mathsf{A}_{j_0} x)}_{n^2 \times 1}.$$

**Definition D.2.** *Let three matrices $A_1, A_2, A_3 \in \mathbb{R}^{n \times d}$ in size $n \times d$. We define $\mathsf{A}_{j_0} \in \mathbb{R}^{n^2 \times d^3}$ be a $n^2 \times d^3$ size sub-block from $\mathsf{A}$ (see as Definition D.1 ). (Recall that $\mathsf{A} = A_1 \otimes A_2 \otimes A_3 \in \mathbb{R}^{n^3 \times d^3}$.)*

*For any index $j_0 \in [n]$, we denote function $\alpha(x)_{j_0} : \mathbb{R}^{d^3} \to \mathbb{R}$ as follows:*

$$\alpha(x)_{j_0} := \langle \underbrace{\exp(\mathsf{A}_{j_0} x)}_{n^2 \times 1}, \underbrace{\mathbf{1}_{n^2}}_{n^2 \times 1} \rangle.$$

**Definition D.3.** *Suppose that $\alpha(x)_{j_0} \in \mathbb{R}$ (see Definition D.2).*

*Recall $\mathsf{K}(x)_{j_0} \in \mathbb{R}^{n^2}$ (see Definition D.1).*

*For a fixed $j_0 \in [n]$, we define function $\mathsf{S}(x)_{j_0} : \mathbb{R}^{d^3} \to \mathbb{R}^{n^2}$ as follows:*

$$\mathsf{S}(x)_{j_0} := \underbrace{\alpha(x)_{j_0}^{-1}}_{\text{scalar}} \underbrace{\mathsf{K}(x)_{j_0}}_{n^2 \times 1}.$$

*We use $\mathsf{S}(x) \in \mathbb{R}^{n \times n^2}$ to denote the matrix where $j_0$-th row is $(\mathsf{S}(x)_{j_0})^\top$. (Note that we can rewrite $\mathsf{S}(x) = D^{-1} \exp(A_1 X (A_2 \otimes A_3)^\top / d) \in \mathbb{R}^{n \times n^2}$ and where $D = \mathrm{diag}(\exp(A_1 X (A_2 \otimes A_3)^\top / d) \mathbf{1}_{n^2})$.)*

**Definition D.4.** *Let $A_3 = A_4 \otimes A_5 \in \mathbb{R}^{n^2 \times d^2}$, where $A_4, A_5, \in \mathbb{R}^{n \times d}$. Let $Y_1, Y_2 \in \mathbb{R}^{d \times d}$. Let $Y = Y_1 \oslash Y_2 \in \mathbb{R}^{d^2 \times d}$ denote the matrix representation of $y \in \mathbb{R}^{d^3}$. For all $i_0 \in [d]$, we define $\mathsf{L}()_{i_0} : \mathbb{R}^{d^3} \to \mathbb{R}^{n^2}$ as follows:*

$$\mathsf{L}(y)_{i_0} := \underbrace{\mathsf{A}_3}_{n^2 \times d^2} \underbrace{Y_{*, i_0}}_{d^2 \times 1}.$$

*Let $\mathsf{L}(y) \in \mathbb{R}^{n^2 \times d}$ matrix where $i_0$ column is $\mathsf{L}(y)_{i_0}$. (Note that we can rewrite $\mathsf{L}(y) = (A_4 \otimes A_5) Y$.)*

We will define $\mathsf{W}$ and $\mathsf{F}$ used in gradient analysis.

**Definition D.5.** *Let $\mathsf{V}(x) \in \mathbb{R}^{n \times d}$ (see Definition D.7). Let $\mathsf{L}(y) \in \mathbb{R}^{n^2 \times d}$ (see Definition D.4).*

*We define $\mathsf{W}(x) \in \mathbb{R}^{n \times n^2}$ to be*

$$\mathsf{W}(x) := \underbrace{\mathsf{V}(x)}_{n \times d} \underbrace{\mathsf{L}(y)^\top}_{d \times n^2}$$

*We denote $\mathsf{W}(x)_{j_0}^\top$ as the $j_0$-th row of $\mathsf{W}(x) \in \mathbb{R}^{n \times n^2}$.*

**Definition D.6.** *For all index $j_0 \in [n]$, let us define $\mathsf{F}(x)_{j_0} \in \mathbb{R}^{n^2}$ to be*

$$\underbrace{\mathsf{F}(x)_{j_0}}_{n^2 \times 1} := \underbrace{(\mathrm{diag}(\mathsf{S}(x)_{j_0}) - \mathsf{S}(x)_{j_0} \mathsf{S}(x)_{j_0}^\top)}_{n^2 \times n^2} \underbrace{\mathsf{W}(x)_{j_0}}_{n^2 \times 1}.$$

*We define $\mathsf{F}(x) \in \mathbb{R}^{n \times n^2}$ in the sense that $\mathsf{F}(x)_{j_0}^\top$ is the $j_0$-th row of $\mathsf{F}(x)$.*

## D.2 DEFINITIONS FOR LOSS FUNCTION

We now present some useful definitions pertaining to $x \in \mathbb{R}^{d^3}$.

**Definition D.7.** *For all $j_0 \in [n]$, we denote $\mathsf{S}(x)_{j_0} \in \mathbb{R}^{n^2}$ as the normalized vector (see Definition D.3). For all $i_0 \in [d]$, we denote $\mathsf{L}(y)_{i_0}$ to be the same in Definition D.4.*

*Consider every $j_0 \in [n]$, every $i_0 \in [d]$. Let us consider $\mathsf{V}(x)_{j_0, i_0} : \mathbb{R}^{d^3} \to \mathbb{R}$ as follows:*

$$\mathsf{V}(x)_{j_0, i_0} := \langle \mathsf{S}(x)_{j_0}, \mathsf{L}(y)_{i_0} \rangle - E_{j_0, i_0},$$

*where $E_{j_0, i_0}$ is the $(j_0, i_0)$-th coordinate of $E \in \mathbb{R}^{n \times d}$ for $j_0 \in [n], i_0 \in [d]$. This is the same as*
$$\underbrace{\mathsf{V}(x)}_{n \times d} = \underbrace{\mathsf{S}(x)}_{n \times n^2} \underbrace{\mathsf{L}(y)}_{n^2 \times d} - \underbrace{E}_{n \times d}.$$

**Definition D.8.** *For all $j_0 \in [n]$, for all $i_0 \in [d]$. We define $\mathsf{Loss}(x)_{j_0, i_0}$ to be $:= 0.5 \mathsf{V}(x)_{j_0, i_0}^2$.*

## D.3 Further information on gradient computation

In this section, we offer detailed analysis to help the computations of gradient and derivative. It is noted that, for the sake of convenience in deriving a closed-form expression for our gradient, we omit the $1/d$ normalization factor in $S$. As this factor merely scales the result, it does not impact the overall computation of these matrices.

**Remark D.9.** *Recall that in Definition 2.8, we consider $X \in \mathbb{R}^{d \times d \times d}$ for gradient computation, which has $d^3$ number of parameters. On the other hand, in Definition 2.9, we have $X = X_1 \cdot (X_2^\top \ominus X_3^\top) \in \mathbb{R}^{d \times d^2}$ which has $3d^2$ number of parameters, which indeed guarantee computation acceleration.*

**Lemma D.10** (The gradient computation for various functions w.r.t. $x_i$). *Let $x \in \mathbb{R}^{d^3}$. Let $j_0 \in [n], i_0 \in [d]$. For all $i \in [d^3]$, we define $\mathsf{A}_{j_0,i} \in \mathbb{R}^{n^2}$ to be the $i$-th column for $\mathsf{A}_{j_0} \in \mathbb{R}^{n^2 \times d^3}$. Recall that $\mathsf{K}(x)_{j_0} \in \mathbb{R}^{n^2}$ is defined in Definitions D.1. The scalar function $\alpha(x)_{j_0} \in \mathbb{R}$ is defined in Definitions D.2 . Column function $\mathsf{S}(x)_{j_0} \in \mathbb{R}^{n^2}$ is defined in Definitions D.3. Scalar function $\mathsf{V}(x)_{j_0,i_0} \in \mathbb{R}$ is defined in Definitions D.7. Scalar function $\mathsf{Loss}(x)_{j_0,i_0} \in \mathbb{R}$ is defined in Definitions D.8.*

*Then, for each $i \in [d^3]$, we have*

- **Part 1.**

$$\frac{\mathrm{d}x}{\mathrm{d}x_i} = e_i$$

- **Part 2.** *For any $j_0 \in [n]$,*

$$\frac{\mathrm{d}\mathsf{A}_{j_0} x}{\mathrm{d}x_i} = \mathsf{A}_{j_0,i}$$

- **Part 3.** *For any $j_0 \in [n]$*

$$\frac{\mathrm{d}\mathsf{K}(x)_{j_0}}{\mathrm{d}x_i} = \mathsf{A}_{j_0,i} \circ \mathsf{K}(x)_{j_0}$$

- **Part 4.** *For any $j_0 \in [n]$,*

$$\frac{\mathrm{d}\alpha(x)_{j_0}}{\mathrm{d}x_i} = \langle \mathsf{A}_{j_0,i}, \mathsf{K}(x)_{j_0} \rangle$$

- **Part 5.** *For any $j_0 \in [n]$,*

$$\frac{\mathrm{d}\mathsf{S}(x)_{j_0}}{\mathrm{d}x_i} = \mathsf{A}_{j_0,i} \circ \mathsf{S}(x)_{j_0} - \langle \mathsf{A}_{j_0,i}, \mathsf{S}(x)_{j_0} \rangle \cdot \mathsf{S}(x)_{j_0}$$

- **Part 6.** *For any $j_0 \in [n]$, for any $i_0 \in [d]$,*

$$\frac{\mathrm{d}\langle \mathsf{S}(x)_{j_0}, \mathsf{L}(y)_{i_0} \rangle}{\mathrm{d}x_i} = \langle \mathsf{L}(y)_{i_0}, \mathsf{A}_{j_0,i} \circ \mathsf{S}(x)_{j_0} \rangle - \langle \mathsf{L}(y)_{i_0}, \mathsf{S}(x)_{j_0} \rangle \cdot \langle \mathsf{A}_{j_0,i}, \mathsf{S}(x)_{j_0} \rangle$$

- **Part 7.** *For any $j_0 \in [n]$, for each $i_0 \in [d]$*

$$\frac{\mathrm{d}\mathsf{V}(x)_{j_0,i_0}}{\mathrm{d}x_i} = \langle \mathsf{A}_{j_0,i} \circ \mathsf{S}(x)_{j_0}, \mathsf{L}(y)_{i_0} \rangle - \langle \mathsf{S}(x)_{j_0}, \mathsf{L}(y)_{i_0} \rangle \cdot \langle \mathsf{A}_{j_0,i}, \mathsf{S}(x)_{j_0} \rangle$$

- **Part 8.** *For any $j_0 \in [n]$, for each $i_0 \in [d]$*

$$\frac{\mathrm{d}\mathsf{Loss}(x)_{j_0,i_0}}{\mathrm{d}x_i} = (\langle \mathsf{L}(y)_{i_0}, \mathsf{A}_{j_0,i} \circ \mathsf{S}(x)_{j_0} \rangle - \langle \mathsf{S}(x)_{j_0}, \mathsf{A}_{j_0,i} \rangle \cdot \langle \mathsf{L}(y)_{i_0}, \mathsf{S}(x)_{j_0} \rangle) \cdot \mathsf{V}(x)_{j_0,i_0}$$

*Proof.* **Proof of Part 1.** We have

$$\frac{\mathrm{d}x}{\mathrm{d}x_i} = \frac{\mathrm{d}[x_1, x_2, \ldots, x_{d^3}]^\top}{\mathrm{d}x_i}$$

$$= e_i$$

where the first step follows from $x$ is a vector, and the second step follows from all coordinates are independent to each other.

**Proof of Part 2.** We have

$$\frac{\mathrm{d}\mathsf{A}_{j_0} x}{\mathrm{d}x_i} = \underbrace{\mathsf{A}_{j_0}}_{n^2 \times d^3} \underbrace{\frac{\mathrm{d}x}{\mathrm{d}x_i}}_{d^3 \times 1}$$

$$= \underbrace{\mathsf{A}_{j_0}}_{n^2 \times d^3} \cdot \underbrace{e_i}_{d^3 \times 1}$$

$$= \underbrace{\mathsf{A}_{j_0,i}}_{n^2 \times 1}$$

where the second step follows from Part 1.

**Proof of Part 3.**

It's easy to show that

$$\underbrace{\frac{\mathrm{d}\mathsf{K}(x)_{j_0}}{\mathrm{d}x_i}}_{n^2 \times 1} = \frac{\mathrm{d}\exp(\mathsf{A}_{j_0} x)}{\mathrm{d}x_i}$$

$$= \exp(\mathsf{A}_{j_0} x) \circ \frac{\mathrm{d}\mathsf{A}_{j_0} x}{\mathrm{d}x_i}$$

$$= \exp(\mathsf{A}_{j_0} x) \circ \mathsf{A}_{j_0,i}$$

$$= \underbrace{\mathsf{K}(x)_{j_0}}_{n^2 \times 1} \circ \underbrace{\mathsf{A}_{j_0,i}}_{n^2 \times 1}$$

where the third step is because of Part 2, the last step follows from definition of $\mathsf{K}(x)_{j_0}$.

**Proof of Part 4.**

To further simplify the writing of proofs, we represent $(x)$ as $(\cdot)$.

It's easy to see that

$$\frac{\mathrm{d}\alpha(\cdot)_{j_0}}{\mathrm{d}x_i} = \frac{\mathrm{d}\langle \mathsf{K}(\cdot)_{j_0}, \mathbf{1}_{n^2} \rangle}{\mathrm{d}x_i}$$

$$= \langle \mathsf{K}(\cdot)_{j_0} \circ \mathsf{A}_{j_0,i}, \mathbf{1}_{n^2} \rangle$$

$$= \langle \mathsf{K}(\cdot)_{j_0}, \mathsf{A}_{j_0,i} \rangle$$

where the first step is due to definition of $\alpha(\cdot)$, the second step is because of Part 3, the third step comes from $\langle a \circ b, \mathbf{1}_{n^2} \rangle = \langle a, b \rangle$.

**Proof of Part 5.**

To further simplify the writing of proofs, we represent $(x)$ as $(\cdot)$.

It's easy to see that

$$\frac{\mathrm{d}\mathsf{S}(\cdot)_{j_0}}{\mathrm{d}x_i} = \frac{\mathrm{d}\alpha(\cdot)_{j_0}^{-1}\mathsf{K}(\cdot)_{j_0}}{\mathrm{d}x_i}$$

$$= \alpha(\cdot)_{j_0}^{-1} \frac{\mathrm{d}\mathsf{K}(\cdot)_{j_0}}{\mathrm{d}x_i} + \left(\frac{\mathrm{d}\alpha(\cdot)_{j_0}^{-1}}{\mathrm{d}x_i}\right)\mathsf{K}(\cdot)_{j_0}$$

For the first term, we have

$$\alpha(\cdot)_{j_0}^{-1} \frac{\mathrm{d}\mathsf{K}(\cdot)_{j_0}}{\mathrm{d}x_i} = \alpha(\cdot)_{j_0}^{-1} \mathsf{K}(\cdot)_{j_0} \circ \mathsf{A}_{j_0,i}$$

$$= \mathsf{S}(\cdot)_{j_0} \circ \mathsf{A}_{j_0,i}$$

where the first step is due to Part 3, the second step is because of definition of $\mathsf{S}(\cdot)$.

For the second term, we have

$$(\frac{\mathrm{d}\alpha(\cdot)_{j_0}^{-1}}{\mathrm{d}x_i})\mathsf{K}(\cdot)_{j_0} = -\alpha(\cdot)_{j_0}^{-2} \frac{\mathrm{d}\alpha(\cdot)_{j_0}}{\mathrm{d}x_i}\mathsf{K}(\cdot)_{j_0}$$

$$= -\alpha(\cdot)_{j_0}^{-2} \cdot \langle \mathsf{K}(\cdot)_{j_0}, \mathsf{A}_{j_0,i} \rangle \cdot \mathsf{K}(\cdot)_{j_0}$$

$$= -\mathsf{S}(\cdot)_{j_0} \cdot \langle \mathsf{S}(\cdot)_{j_0}, \mathsf{A}_{j_0,i} \rangle$$

where the first step is from simple calculus, the second step is from Part 4, and the third step is due to the definition of $\mathsf{S}(\cdot)_{j_0}$.

By applying all of the above, we have

$$\frac{\mathrm{d}\mathsf{S}(\cdot)_{j_0}}{\mathrm{d}x_i} = \mathsf{S}(\cdot)_{j_0} \circ \mathsf{A}_{j_0,i} - \mathsf{S}(\cdot)_{j_0} \cdot \langle \mathsf{S}(\cdot)_{j_0}, \mathsf{A}_{j_0,i} \rangle$$

**Proof of Part 6.** From Part 5, clearly this holds.

**Proof of Part 7.**

To further simplify the writing of proofs, we represent $(x)$ as $(\cdot)$.

From definition of $\mathsf{V}$ in Definition D.7, it holds that

$$\mathsf{V}(\cdot)_{j_0,i_0} := \langle \mathsf{S}(\cdot)_{j_0}, \mathsf{L}(y)_{i_0} \rangle - E_{j_0,i_0} \tag{2}$$

Thus it holds that

$$\frac{\mathrm{d}\mathsf{V}(\cdot)_{j_0,i_0}}{\mathrm{d}x_i} = \frac{\mathrm{d}(\langle \mathsf{S}(\cdot)_{j_0}, \mathsf{L}(y)_{i_0} \rangle - E_{j_0,i_0})}{\mathrm{d}x_i}$$

$$= \frac{\mathrm{d}\langle \mathsf{S}(\cdot)_{j_0}, \mathsf{L}(y)_{i_0} \rangle}{\mathrm{d}x_i}$$

$$= \langle \mathsf{S}(\cdot)_{j_0} \circ \mathsf{A}_{j_0,i}, \mathsf{L}(y)_{i_0} \rangle - \langle \mathsf{S}(\cdot)_{j_0}, \mathsf{L}(y)_{i_0} \rangle \cdot \langle \mathsf{S}(\cdot)_{j_0}, \mathsf{A}_{j_0,i} \rangle,$$

where the first step comes from Eq. (2), the second step follows from $\frac{\mathrm{d}E_{j_0,i_0}}{\mathrm{d}x_i} = 0$, and the last step is due to **Part 6**.

**Proof of Part 8.**

To further simplify the writing of proofs, we represent $(x)$ as $(\cdot)$.

From definition of $\mathsf{Loss}(\cdot)$ (see Definition D.8), it holds that

$$\mathsf{Loss}(\cdot)_{j_0,i_0} = 0.5\mathsf{V}(\cdot)_{j_0,i_0}^2 \tag{3}$$

Thus, we have

$$\frac{\mathrm{d}\mathsf{Loss}(\cdot)_{j_0,i_0}}{\mathrm{d}x_i} = \frac{\mathrm{d}(0.5\mathsf{V}(\cdot)_{j_0,i_0}^2)}{\mathrm{d}x_i}$$

$$= \mathsf{V}(\cdot)_{j_0,i_0} \frac{\mathrm{d}\mathsf{V}(\cdot)}{\mathrm{d}x_i}$$

$$= \mathsf{V}(\cdot)_{j_0,i_0} \cdot (\langle \mathsf{S}(\cdot)_{j_0} \circ \mathsf{A}_{j_0,i}, \mathsf{L}(y)_{i_0} \rangle - \langle \mathsf{S}(\cdot)_{j_0}, \mathsf{L}(y)_{i_0} \rangle \cdot \langle \mathsf{S}(\cdot)_{j_0}, \mathsf{A}_{j_0,i} \rangle),$$

where the 1st step comes from the Eq. (3), the second step follows from the chain rule, and the last step is because of **Part 7**.

$\square$

# E TENSOR ATTENTION EXACT GRADIENT COMPUTATION TIME COMPLEXITY

Section E.1 demonstrates how to calculate $\mathsf{S}$ ($1/d$ factor is still ignored) and $\mathsf{L}$. Section E.2 explains the straightforward method for calculating $\mathsf{V}$. Section E.3 and Section E.4 define $\mathsf{F}$ and $\mathsf{W}$, and demonstrate their computations. Section E.5 presents a more elegant way to express the gradient. Finally, Section E.6 combines all these elements and determine the overall time complexity of our algorithm.

## E.1 TIME COMPLEXITY TO GET $\mathsf{S}$ AND $\mathsf{L}$

**Remark E.1.** *Note that* $\mathcal{T}_{\mathrm{mat}}(n, d^2, n^2) \geq \Omega(n^3)$.

Now we will show the time complexity for computing $\mathsf{S}$ and $\mathsf{L}$.

**Lemma E.2** (Computing $\mathsf{S}$ and $\mathsf{L}$). *If the following conditions hold*

- *Let* $\mathsf{S}(x) \in \mathbb{R}^{n \times n^2}$ *(see Definition D.3)*

- *Let* $\mathsf{L}(y) \in \mathbb{R}^{n^2 \times d}$ *(see Definition D.4)*

*Then, we have*

- *the time complexity of* $\mathsf{S}(x)$ *is* $\mathcal{T}_{\mathrm{mat}}(n, d^2, n^2) + \mathcal{T}_{\mathrm{mat}}(n, d, d^2)$

- *the time complexity of* $\mathsf{L}(y)$ *is* $\mathcal{T}_{\mathrm{mat}}(n^2, d^2, d)$

*Proof.* Note that

$$\mathsf{S}(x) = \underbrace{D^{-1}}_{n \times n} \exp(\underbrace{A_1}_{n \times d} \underbrace{X}_{d \times d^2} \underbrace{(A_2 \otimes A_3)^\top}_{d^2 \times n^2})$$

and

$$D = \mathrm{diag}(\exp(A_1 X (A_2 \otimes A_3)^\top) \mathbf{1}_{n^2})$$

We firstly compute $\exp(A_1 X (A_2 \otimes A_3)^\top)$, this takes time of

- $\underbrace{A_1}_{n \times d} \underbrace{X}_{d \times d^2}$ takes $\mathcal{T}_{\mathrm{mat}}(n, d, d^2)$

- Computing $A_2 \otimes A_3$ takes $O(n^2 d^2)$ time

- Computing $A_1 X \cdot (A_2 \otimes A_3)^\top$ takes $\mathcal{T}_{\mathrm{mat}}(n, d^2, n^2)$ time

The overall time complexity of above three parts is dominated by

$$\mathcal{T}_{\mathrm{mat}}(n, d, d^2) + O(d^2 n^2) + \mathcal{T}_{\mathrm{mat}}(n, d^2, n^2) = \mathcal{T}_{\mathrm{mat}}(n, d, d^2) + \mathcal{T}_{\mathrm{mat}}(n, d^2, n^2)$$

Therefore, computing $D$ takes $O(n^3)$ time.

Computing $D^{-1} \exp(A_1 X (A_2 \otimes A_3)^\top)$ requires $O(n^3)$ time.

Therefore, the overall time complexity is

$$\mathcal{T}_{\mathrm{mat}}(n, d, d^2) + \mathcal{T}_{\mathrm{mat}}(n, d^2, n^2)$$

It is noted that computing $\mathsf{L}(y) = \underbrace{A_3}_{n^2 \times d^2} \underbrace{Y}_{d^2 \times d}$ takes time of $\mathcal{T}_{\mathrm{mat}}(n^2, d^2, d)$.

Thus, we complete the proof. $\square$

## E.2 TIME COMPLEXITY TO GET $\mathsf{V}$

We will explain the calculation of $\mathsf{V}$.

**Lemma E.3** (Computing $\mathsf{V}$). *If the following conditions hold*

- *Let $E \in \mathbb{R}^{n \times d}$*

- *Let $\mathsf{S}(x) \in \mathbb{R}^{n \times n^2}$.*

- *Let $\mathsf{L}(y) \in \mathbb{R}^{n^2 \times d}$.*

*Then one can get $\mathsf{V}(x) \in \mathbb{R}^{n \times d}$ in $O(\mathcal{T}_{\mathrm{mat}}(n, n^2, d))$ time.*

*Proof.* Based on the definition of $\mathsf{V}(x) \in \mathbb{R}^{n \times d}$ which is

$$\mathsf{V}(x) = \underbrace{\mathsf{S}(x)}_{n \times n^2} \underbrace{\mathsf{L}(y)}_{n^2 \times d} - \underbrace{E}_{n \times d}$$

It is easy to see that we can compute $\mathsf{S}(x)\mathsf{L}(y)$ in time $\mathcal{T}_{\mathrm{mat}}(n, n^2, d)$, and $\mathsf{S}(x)\mathsf{L}(y) - E$ in time $O(nd)$.

Therefore, overall running time is

$$\mathcal{T}_{\mathrm{mat}}(n, n^2, d) + O(nd) = O(\mathcal{T}_{\mathrm{mat}}(n, n^2, d)).$$

$\square$

## E.3 TIME COMPLEXITY TO GET $\mathsf{W}$

We will explain how to calculate $\mathsf{W}$.

**Lemma E.4.** *If the below holds that*

- *Let $\mathsf{V}(x) \in \mathbb{R}^{n \times d}$*

- *Let $\mathsf{L}(y) \in \mathbb{R}^{n^2 \times d}$*

*Then, computing $\mathsf{W}(x)$ takes time of $O(\mathcal{T}_{\mathrm{mat}}(n, d, n^2))$.*

*Proof.* Let use recall that $\mathsf{W}(x) = \mathsf{V}(x)\mathsf{L}(y)^{\top}$. This need time of $\mathcal{T}_{\mathrm{mat}}(n, d, n^2)$ to compute. $\square$

## E.4 TIME COMPLEXITY TO GET $\mathsf{F}$

We can show how to construct $\mathsf{F}$.

**Lemma E.5.** *If the following conditions hold*

- *Let $\mathsf{S}(x) \in \mathbb{R}^{n \times n^2}$*

- *Let $\mathsf{W}(x) \in \mathbb{R}^{n \times n^2}$*

*Then, computing takes time of $\mathsf{F}(x)$ in $O(n^3)$.*

*Proof.* For every $j_0 \in [n]$, it follows that $\mathsf{F}(x)_{j_0} \in \mathbb{R}^{n^2}$ can be computed in $O(n^2)$, given that $\mathrm{diag}(\mathsf{S}(x)_{j_0})$ is a diagonal matrix and $\mathsf{S}(x)_{j_0}\mathsf{S}(x)_{j_0}^{\top}$ is a rank-one matrix. Consequently, constructing the matrix $\mathsf{F}(x) \in \mathbb{R}^{n \times n^2}$ takes a total time of $n \times O(n^2) = O(n^3)$. $\square$

### E.5 CLOSED FORM OF GRADIENT

We will give the closed form the gradient of the loss function.

**Lemma E.6** (Closed form of gradient, formal version of Lemma 3.1). *Let us define functions* $\mathsf{S}(x) \in \mathbb{R}^{n \times n^2}$, $\mathsf{V}(x) \in \mathbb{R}^{n \times d}$, $\mathsf{L}(y) \in \mathbb{R}^{n^2 \times d}$, $\mathsf{W}(x) \in \mathbb{R}^{n \times n^2}$ *and* $\mathsf{F}(x) \in \mathbb{R}^{n \times n^2}$ *(see Definitions D.3, D.7, D.4, D.5 and D.6 respectively). Suppose three matrices* $A_1, A_2, A_3 \in \mathbb{R}^{n \times d}$ *are given. We define* $\mathsf{A} = A_1 \otimes A_2 \otimes A_3$. *Let* $\mathsf{Loss}(x)$ *and* $\mathsf{Loss}(x)_{j_0, i_0}$ *be defined as Definition 2.8 and D.8. Then, we can show that*

$$\frac{\mathrm{d}\mathsf{Loss}(x)}{\mathrm{d}x} = \mathrm{vec}(A_1^\top \mathsf{F}(x)(A_2 \otimes A_3)) \in \mathbb{R}^{d^3}.$$

*Proof.* From the Lemma statement and Lemma D.10 Part 8, we have

$$\frac{\mathrm{d}\mathsf{Loss}(x,y)_{j_0, i_0}}{\mathrm{d}x_i} = \mathsf{V}(x,y)_{j_0, i_0} \cdot (\langle \mathsf{S}(x)_{j_0} \circ \mathsf{A}_{j_0, i}, \mathsf{L}(y)_{i_0} \rangle - \langle \mathsf{S}(x)_{j_0}, \mathsf{L}(y)_{i_0} \rangle \cdot \langle \mathsf{S}(x)_{j_0}, \mathsf{A}_{j_0, i} \rangle) \tag{4}$$

We know that for all $a, b \in \mathbb{R}^n$, we have $\mathrm{diag}(a) \cdot b = \mathrm{diag}(b) \cdot a = a \circ b = b \circ a$. Then, we have

$$\langle \mathsf{S}(x)_{j_0} \circ \mathsf{A}_{j_0, i}, \mathsf{L}(y)_{i_0} \rangle = (\mathrm{diag}(\mathsf{S}(x)_{j_0}) \mathsf{A}_{j_0, i})^\top \mathsf{L}(y)_{i_0} = \mathsf{A}_{j_0, i}^\top \mathrm{diag}(\mathsf{S}(x)_{j_0}) \mathsf{L}(y)_{i_0}$$

and

$$\langle \mathsf{S}(x)_{j_0}, \mathsf{L}(y)_{i_0} \rangle \cdot \langle \mathsf{S}(x)_{j_0}, \mathsf{A}_{j_0, i} \rangle = \mathsf{A}_{j_0, i}^\top \mathsf{S}(x)_{j_0} \mathsf{S}(x)_{j_0}^\top \mathsf{L}(y)_{i_0}$$

Therefore, Eq. (4) becomes

$$\frac{\mathrm{d}\mathsf{Loss}(x)_{j_0, i_0}}{\mathrm{d}x_i} = \mathsf{V}(x,y)_{j_0, i_0} \cdot (\mathsf{A}_{j_0, i}^\top \mathrm{diag}(\mathsf{S}(x)_{j_0}) \mathsf{L}(y)_{i_0} - \mathsf{A}_{j_0, i}^\top \mathsf{S}(x)_{j_0} \mathsf{S}(x)_{j_0}^\top \mathsf{L}(y)_{i_0})$$

$$= \mathsf{V}(x,y)_{j_0, i_0} \cdot \mathsf{A}_{j_0, i}^\top (\mathrm{diag}(\mathsf{S}(x)_{j_0}) - \mathsf{S}(x)_{j_0} \mathsf{S}(x)_{j_0}^\top) \mathsf{L}(y)_{i_0}, \tag{5}$$

where the second step is due to basic algebra.

Note that we defined $\mathsf{W}(x)_{j_0}$ in Definition D.5.

$$\mathsf{W}(x)_{j_0} := \sum_{i_0 = 1}^{d} \mathsf{V}(x)_{j_0, i_0} \mathsf{L}(y)_{i_0}. \tag{6}$$

Also, we defined $\mathsf{F}(x)_{j_0} \in \mathbb{R}^{n^2}$ in Definition D.6,

$$\mathsf{F}(x)_{j_0} := (\mathrm{diag}(\mathsf{S}(x)_{j_0}) - \mathsf{S}(x)_{j_0} \mathsf{S}(x)_{j_0}^\top) \mathsf{W}(x)_{j_0}. \tag{7}$$

We can show

$$\frac{\mathrm{d}\mathsf{Loss}(x)}{\mathrm{d}x}$$

$$= \sum_{j_0 = 1}^{n} \sum_{i_0 = 1}^{d} \frac{\mathrm{d}\mathsf{Loss}(x)_{j_0, i_0}}{\mathrm{d}x}$$

$$= \sum_{j_0 = 1}^{n} \sum_{i_0 = 1}^{d} \underbrace{\mathsf{V}(x)_{j_0, i_0}}_{\text{scalar}} \cdot \underbrace{\mathsf{A}_{j_0}^\top}_{d^3 \times n^2} \underbrace{(\mathrm{diag}(\mathsf{S}(x)_{j_0}) - \mathsf{S}(x)_{j_0} \mathsf{S}(x)_{j_0}^\top)}_{n^2 \times n^2} \underbrace{\mathsf{L}(y)_{i_0}}_{n^2 \times 1}$$

$$= \sum_{j_0 = 1}^{n} \mathsf{A}_{j_0}^\top (\mathrm{diag}(\mathsf{S}(x)_{j_0}) - \mathsf{S}(x)_{j_0} \mathsf{S}(x)_{j_0}^\top) \mathsf{W}(x)_{j_0}$$

$$= \sum_{j_0 = 1}^{n} \mathsf{A}_{j_0}^\top \mathsf{F}(x)_{j_0}$$

$$= A^\top \mathrm{vec}(\mathsf{F}(x))$$

$$= \mathrm{vec}(A_1^\top \mathsf{F}(x)(A_2 \otimes A_3)) \in \mathbb{R}^{d^3}$$

where the first step comes from Definition 2.8, the second step is due to Eq. (5), the third step is because of Eq. (6), the fourth step is due to Eq. (7), the fifth step utilize the notation of $\mathrm{vec}(\cdot)$, and the last step follows from Fact C.16.

$\square$

### E.6 PUTTING ALL TOGETHER

We now show the overall running time of computing the gradient.

**Theorem E.7** (Tensor attention gradient computation, formal version of Theorem 3.3 ). *If we have the following conditions*

- *Suppose that we have input fixed matrices $A_1, A_2, A_3, A_4, A_5, E \in \mathbb{R}^{n \times d}$.*

- *We denote $X \in \mathbb{R}^{d \times d^2}$ and $Y \in \mathbb{R}^{d^2 \times d}$ as matrix variables (gradient is computed w.r.t. $X$ )*

  - *For simplicity of calculation, we utilize vector variables $x \in \mathbb{R}^{d^3 \times 1}$ and $y \in \mathbb{R}^{d^3 \times 1}$, i.e., $\mathrm{vec}(X) = x$.*
  - *For simplicity of calculation, we use tensor variables $\mathsf{X} \in \mathbb{R}^{d \times d \times d}$ and $\mathsf{Y} \in \mathbb{R}^{d \times d \times d}$*

- *Let $g = \frac{\mathrm{d}\mathsf{Loss}(X)}{\mathrm{d}X} \in \mathbb{R}^{d \times d^2}$ (see $\mathsf{Loss}(X)$ in Definition 2.8)*

*Then it's plain to see that we can compute gradient $g \in \mathbb{R}^{d \times d^2}$ in $\mathcal{T}_{\mathrm{mat}}(n, d^2, n^2)$ time.*

*Proof.* Step 1. We compute $\mathsf{S}(x)$ and $\mathsf{L}(y)$. According to Lemma E.2, this takes $O(\mathcal{T}_{\mathrm{mat}}(n, d^2, n^2) + \mathcal{T}_{\mathrm{mat}}(n, d, d^2))$ time.

Step 2. We compute $\mathsf{V}(x)$. According to Lemma E.3, this takes $O(\mathcal{T}_{\mathrm{mat}}(n, n^2, d))$ time.

Step 3. We compute $\mathsf{W}(x)$. According to Lemma E.4, this takes $O(\mathcal{T}_{\mathrm{mat}}(n, d, n^2))$ time.

Step 4. We compute $\mathsf{F}(x)$. According to Lemma E.5, this takes $O(n^3)$ time.

Step 5. From Lemma E.6, the gradient is give by $\mathrm{vec}(A_1^\top \mathsf{F}(x)(A_2 \otimes A_3))$. We know that $A_1^\top \in \mathbb{R}^{d \times n}$, $\mathsf{F}(x) \in \mathbb{R}^{n \times n^2}$, and $A_2 \otimes A_3 \in \mathbb{R}^{n^2 \times d^2}$, it can be calculated in $O(\mathcal{T}_{\mathrm{mat}}(d, n, d^2) + \mathcal{T}_{\mathrm{mat}}(n, n^2, d^2))$ time.

Thus, the overall running time complexity for computing the gradient is $O(\mathcal{T}_{\mathrm{mat}}(n, d^2, n^2) + \mathcal{T}_{\mathrm{mat}}(n, d, d^2))$. $\square$

## F RUNNING ACCELERATION VIA POLYNOMIAL METHOD

Remember that in the preceding section, for simplicity in the computations of the gradient, we didn't consider the $d$ factor in $\mathsf{S}$. This factor does not affect the time complexity in our algorithms as it merely acts as a rescaling factor. We will now retake the $1/d$ in $\mathsf{S}$ factor into consideration to utilize the tools from previous work (Alman & Song, 2023).

In Section F.1, we demonstrate how to create a low-rank representation for $\mathsf{S}$ efficiently and explicitly. In Section F.2, we show how to make a low-rank construction for $\mathsf{V}(x)$. In Sections F.3, F.4, and F.5, we present low-rank representations for $\mathsf{W}(x)$, $\mathsf{F}_a(x)$, and $\mathsf{F}_b(x)$, respectively. Finally, in Section F.6, we will consolidate all these elements to prove our final algorithmic result.

### F.1 FAST COMPUTATION OF $\mathsf{S}$

Using the polynomial method results in (Alman & Song, 2023; 2024b), we have the following low-rank representation results.

**Lemma F.1.** *For any $B = o(\sqrt[3]{\log n})$, we have $k_1 = n^{o(1)}$ such that: Let $A_1, A_2, A_3 \in \mathbb{R}^{n \times d}$, $X_1, X_2, X_3 \in \mathbb{R}^{d \times d}$ and $X = X_1 \cdot (X_2^\top \ominus X_3^\top) \in \mathbb{R}^{d \times d^2}$. Assume that each number in $\mathsf{S}(x)$ can be written using $O(\log n)$ bits. It holds that $\max\{\|A_1 X_1\|_\infty, \|A_2 X_2\|_\infty, \|A_3 X_3\|_\infty\} \le B$, then there are three matrices $U_1, V_1, W_1 \in \mathbb{R}^{n \times k_1}$ such that $\|U_1 (V_1 \oslash W_1)^\top - \mathsf{S}(x)\|_\infty \le \epsilon/\operatorname{poly}(n)$. Here $\mathsf{S}(x) = D^{-1} \exp(A_1 X (A_2 \otimes A_3)^\top / d) \in \mathbb{R}^{n \times n^2}$ and we define $D = \operatorname{diag}(\exp(A_1 X (A_2 \otimes A_3)^\top / d) \mathbf{1}_{n^2})$. Moreover, these matrices $U_1, V_1, W_1$ can be created explicitly in $n^{1+o(1)}$ time.*

*Proof.* We have

$$(X_2^\top \ominus X_3^\top) \cdot (A_2 \otimes A_3)^\top = ((A_2 \otimes A_3) \cdot (X_2^\top \ominus X_3^\top)^\top)^\top$$
$$= ((A_2 \otimes A_3) \cdot (X_2 \oslash X_3))^\top$$
$$= ((A_2 \cdot X_2) \oslash (A_3 \cdot X_3))^\top,$$

where the first step is due to simple algebra, the second step comes from Fact C.4, and the last step follows Fact C.7.

Thus, we can rewrite $\mathsf{S}(x) = D^{-1} \exp(Q(K_1 \oslash K_2)^\top / d) \in \mathbb{R}^{n \times n^2}$ and we define $D = \operatorname{diag}(\exp(Q(K_1 \oslash K_2)^\top / d) \mathbf{1}_{n^2})$, where $Q = A_1 X_1, K_1 = A_2 X_2, K_2 = A_3 X_3$.

More explicitly, we have

$$Q(K_1 \oslash K_2)^\top = A_1 X_1 (A_2 X_2 \oslash A_3 X_3)^\top$$
$$= A_1 X_1 (X_2^\top \ominus X_3^\top) \cdot (A_2 \otimes A_3)^\top$$
$$= A_1 X (A_2 \otimes A_3)^\top,$$

where the 1st step is due to $Q = A_1 X_1, K_1 = A_2 X_2, K_2 = A_3 X_3$, the 2nd step is because of the identity in the beginning of the proof, and the 3rd step follows from $X = X_1 (X_2^\top \ominus X_3^\top)$.

Thus, we finish the proof by applying Lemma 4.1. $\square$

### F.2 Fast computation of $\mathsf{V}$

We will explain how to obtain the low rank representation of $\mathsf{V}(x)$.

**Lemma F.2.** *We assume conditions the same as Lemma F.1. Let $d = O(\log n)$ and $k_1 = n^{o(1)}$. We also assume that we can write each number in $E \in \mathbb{R}^{n \times d}$ and $\mathsf{L}(y) \in \mathbb{R}^{n^2 \times d}$ using $O(\log n)$ bits. Let $\mathsf{V}(x) \in \mathbb{R}^{n \times d}$ (see Definition D.7). Then, there are three matrices $U_1, V_1, W_1 \in \mathbb{R}^{n \times k_1}$ we have $\|U_1 (V_1 \oslash W_1)^\top \mathsf{L}(y) - E - \mathsf{V}(x)\|_\infty \le \epsilon/\operatorname{poly}(n)$, where $V_1 \oslash W_1 \in \mathbb{R}^{n^2 \times k_1}$. Moreover, we can construct these matrices $U_1, V_1, W_1$ in $n^{1+o(1)}$ time.*

*Proof.* Let $U_1, V_1, W_1$ be the matrices in Lemma F.1. We can show that

$$\|U_1 (V_1 \oslash W_1)^\top \mathsf{L}(y) - E - \mathsf{V}(x)\|_\infty = \|U_1 (V_1 \oslash W_1)^\top \mathsf{L}(y) - E - \mathsf{S}(x)\mathsf{L}(y) + E\|_\infty$$
$$= \|(U_1 (V_1 \oslash W_1)^\top - \mathsf{S}(x))\mathsf{L}(y)\|_\infty$$
$$\le \epsilon/\operatorname{poly}(n)$$

where the 1st step is due to $\mathsf{V}(x) = \mathsf{S}(x)\mathsf{L}(y) - E$, the 2nd step comes from basic algebra, and 3rd step is due to Lemma F.1 and each number in $\mathsf{L}(y) \in \mathbb{R}^{n^2 \times d}$ can be written using $O(\log n)$.

$\square$

### F.3 Fast computation of $\mathsf{W}$

We will explain how to obtain the low rank representation of $\mathsf{W}(x)$.

**Lemma F.3.** *Assume the same condition as Lemma F.2. Let $k_2 = n^{o(1)}$. We define $\mathsf{V}(x) \in \mathbb{R}^{n \times d}$ (see Definition D.7). We define $\mathsf{L}(y) \in \mathbb{R}^{n^2 \times d}$ (see Definition D.4). Let $\mathsf{W}(x) := \mathsf{V}(x)\mathsf{L}(y)^\top \in \mathbb{R}^{n \times n^2}$ be defined in Definition D.5. There are three matrices $U_2, V_2, W_2 \in \mathbb{R}^{n \times k_2}$ such that $\|U_2 (V_2 \oslash W_2)^\top - \mathsf{W}(x)\|_\infty \le \epsilon/\operatorname{poly}(n)$. We can construct the matrices $U_2, V_2, W_2$ in $n^{1+o(1)}$ time.*

*Proof.* For $\mathsf{W}(x)$, we define its approximation as $\widetilde{\mathsf{W}}(x)$.

According to Lemma F.2, we find a good approximation $U_1(V_1 \oslash W_1)^\top \mathsf{L}(y) - E$ of $\mathsf{V}(x)$, where $k_1 = n^{o(1)}$ and $U_1, V_1, W_1 \in \mathbb{R}^{n \times k_1}$.

Now we turn $\widetilde{\mathsf{W}}(x)$ into low-rank representation

$$\widetilde{\mathsf{W}}(x) = \underbrace{(U_1(V_1 \oslash W_1)^\top \mathsf{L}(y) - E)}_{n \times d} \underbrace{\mathsf{L}(y)^\top}_{d \times n^2}$$

$$= \underbrace{(U_1(V_1 \oslash W_1)^\top \mathsf{L}(y) - E)}_{n \times d} \underbrace{((A_4 \otimes A_5) \cdot (Y_1 \oslash Y_2))^\top}_{d \times n^2}$$

$$= \underbrace{(U_1(V_1 \oslash W_1)^\top \mathsf{L}(y) - E)}_{n \times d} (\underbrace{(A_4 \cdot Y_1)}_{n \times d} \oslash \underbrace{(A_5 \cdot Y_2)}_{n \times d})^\top,$$

where the 1st step is because that $U_1(V_1 \oslash W_1)^\top \mathsf{L}(y) - E$ is a good approximation to $\mathsf{V}(x)$, the 2nd step comes from definition of $\mathsf{L}(y)$ (see Definition D.4), the last step is due to Fact C.7.

Thus, we let $U_2 = U_1(V_1 \oslash W_1)^\top \mathsf{L}(y) - E$, $V_2 = A_4 \cdot Y_1$ and $W_2 = A_5 \cdot Y_2$, which only takes $n^{1+o(1)}$ time. (We remark that, if we use naive way to compute $U_2$ that it takes $\Omega(n^2)$, however using Lemma C.13 can beat $O(n^2)$ time.) We can explicitly construct $U_2, V_2, W_2 \in \mathbb{R}^{n \times k_2}$ where $k_2 \leq \max\{d, k_1\} + d = n^{o(1)}$. (Here the reason is $k_1 = n^{o(1)}$ and $d = n^{o(1)}$)

For controlling the error, we can show

$$\|\widetilde{\mathsf{W}}(x) - \mathsf{W}(x)\|_\infty = \|(U_1(V_1 \oslash W_1)^\top \mathsf{L}(y) - E)\mathsf{L}(y)^\top - \mathsf{V}(x)\mathsf{L}(y)^\top\|_\infty$$

$$\leq d \cdot \|\mathsf{L}(y)\|_\infty \cdot \|U_1(V_1 \oslash W_1)^\top \mathsf{L}(y) - E - \mathsf{V}(x)\|_\infty$$

$$\leq \epsilon/\operatorname{poly}(n),$$

where the first step follows from the definition of $\widetilde{\mathsf{W}}(x), \mathsf{W}(x)$, the second step follows from $\|ab^\top\|_\infty \leq d \cdot \|a\|_\infty \cdot \|b\|_\infty$ for length $d$ vectors $a, b$, and the last step follows Lemma F.2.

Thus, we complete the proof. $\square$

### F.4 FAST COMPUTATION OF $\mathsf{F}_a$: KEY STEP

**Definition F.4.** *Let $\mathsf{S}(x) \in \mathbb{R}^{n \times n^2}$ (see Definition D.3). Let $\mathsf{W}(x) \in \mathbb{R}^{n \times n^2}$ (see Definition D.5). Then, we define*

$$\mathsf{F}_a(x) := \mathsf{S}(x) \circ \mathsf{W}(x) \in \mathbb{R}^{n \times n^2}.$$

We will explain how to obtain the low-rank representation of $\mathsf{F}_a(x)$.

**Lemma F.5.** *Let $k_1 = n^{o(1)}$, $k_2 = n^{o(1)}$, $k_3 = n^{o(1)}$. We assume $U_1, V_1, W_1 \in \mathbb{R}^{n \times k_1}$ approximates the $\mathsf{S}(x) \in \mathbb{R}^{n \times n^2}$ satisfying $\|U_1(V_1 \oslash W_1)^\top - \mathsf{S}(x)\|_\infty \leq \epsilon/\operatorname{poly}(n)$. Let us assume that $U_2, V_2, W_2 \in \mathbb{R}^{n \times k_2}$ approximates the $\mathsf{W}(x) \in \mathbb{R}^{n \times n^2}$ satisfying $\|U_2(V_2 \oslash W_2)^\top - \mathsf{W}(x)\|_\infty \leq \epsilon/\operatorname{poly}(n)$. We assume that each number in $\mathsf{S}(x)$ and $\mathsf{W}(x)$ can be written using $O(\log n)$ bits. Let $\mathsf{F}_a(x) := \mathsf{S}(x) \circ \mathsf{W}(x) \in \mathbb{R}^{n \times n^2}$ be defined in Definition F.4. Then there are matrices $U_3, V_3, W_3 \in \mathbb{R}^{n \times k_3}$ such that $\|U_3(V_3 \oslash W_3)^\top - \mathsf{F}_a(x)\|_\infty \leq \epsilon/\operatorname{poly}(n)$. We can construct the matrices $U_3, V_3, W_3$ in $n^{1+o(1)}$ time.*

*Proof.* If we choose $U_3 = U_1 \ominus U_2 \in \mathbb{R}^{n \times k_1 k_2}$ and $V_3 = V_1 \ominus V_2 \in \mathbb{R}^{n \times k_1 k_2}$, $W_3 = W_1 \ominus W_2 \in \mathbb{R}^{n \times k_1 k_2}$, this need $n^{1+o(1)}$ time to compute.

For further simplicity of proofs, we call $\widetilde{\mathsf{S}}(x) = U_1(V_1 \oslash W_1)^\top$ and $\widetilde{\mathsf{W}}(x) = U_2(V_2 \oslash W_2)^\top$.

According to Lemma C.13, we can show

$$\|U_3(V_3 \oslash W_3)^\top - \mathsf{F}_a(x)\|_\infty = \|U_3(V_3 \oslash W_3)^\top - \mathsf{S}(x) \circ \mathsf{W}(x)\|_\infty$$

$$= \|(U_1 \ominus U_2)((V_1 \ominus V_2) \oslash (W_1 \ominus W_2))^\top - \mathsf{S}(x) \circ \mathsf{W}(x)\|_\infty$$

$$= \|(U_1(V_1 \oslash W_1)^\top) \circ (U_2(V_2 \oslash W_2)^\top) - \mathsf{S}(x) \circ \mathsf{W}(x)\|_\infty$$

$$= \|\widetilde{\mathsf{S}}(x) \circ \widetilde{\mathsf{W}}(x) - \mathsf{S}(x) \circ \mathsf{W}(x)\|_\infty$$

$$= \|\widetilde{\mathsf{S}}(x) \circ \widetilde{\mathsf{W}}(x) - \widetilde{\mathsf{S}}(x) \circ \mathsf{W}(x) + \widetilde{\mathsf{S}}(x) \circ \mathsf{W}(x) - \mathsf{S}(x) \circ \mathsf{W}(x)\|_\infty$$

$$\leq \|\widetilde{\mathsf{S}}(x) \circ \widetilde{\mathsf{W}}(x) - \widetilde{\mathsf{S}}(x) \circ \mathsf{W}(x)\|_\infty + \|\widetilde{\mathsf{S}}(x) \circ \mathsf{W}(x) - \mathsf{S}(x) \circ \mathsf{W}(x)\|_\infty$$

$$\leq \epsilon/\operatorname{poly}(n)$$

where the first step is due to the definition of $\mathsf{F}_a(x)$, the second step is because of the definition of $U_3, V_3, W_3$, the third step is due to Fact C.8, the fourth step follows from the definition of $\widetilde{\mathsf{S}}(x)$ and $\widetilde{\mathsf{W}}(x)$, the fifth step is because of basic algebra, the sixth step comes from triangle inequality, and the last step is because bounded entries (we can write each number in $\mathsf{S}(x)$ and $\mathsf{W}(x)$ using $O(\log n)$ bits) and Lemma assumptions that $\|\widetilde{\mathsf{S}}(x) - \mathsf{S}(x)\|_\infty \leq \epsilon/\operatorname{poly}(n)$ and $\|\widetilde{\mathsf{W}}(x) - \mathsf{W}(x)\|_\infty \leq \epsilon/\operatorname{poly}(n)$

$\square$

## F.5 FAST COMPUTATION OF $\mathsf{F}_b$: KEY STEP

**Definition F.6.** *Let $\mathsf{S}(x) \in \mathbb{R}^{n \times n^2}$ (see Definition D.3). Let $\mathsf{W}(x) \in \mathbb{R}^{n \times n^2}$ (see Definition D.5). Then, we define $\mathsf{F}_b(x) \in \mathbb{R}^{n \times n^2}$ whose $j_0$-th column*

$$\mathsf{F}_b(x)_{j_0} = \mathsf{S}(x)_{j_0} \mathsf{S}(x)_{j_0}^\top \mathsf{W}(x)_{j_0},$$

*for each $j_0 \in [n]$.*

We will explain how to obtain the low rank representation of $\mathsf{F}_b(x)$.

**Lemma F.7.** *Let $k_1 = n^{o(1)}$, $k_2 = n^{o(1)}$, $k_4 = n^{o(1)}$. Let us assume that $U_1, V_1, W_1 \in \mathbb{R}^{n \times k_1}$ approximates the $\mathsf{S}(x) \in \mathbb{R}^{n \times n^2}$ satisfying $\|U_1(V_1 \oslash W_1)^\top - \mathsf{S}(x)\|_\infty \leq \epsilon/\operatorname{poly}(n)$. We assume $U_2, V_2, W_2 \in \mathbb{R}^{n \times k_2}$ approximates the $\mathsf{W}(x) \in \mathbb{R}^{n \times n^2}$ satisfying $\|U_2(V_2 \oslash W_2)^\top - \mathsf{W}(x)\|_\infty \leq \epsilon/\operatorname{poly}(n)$. Assume that we can write each number in $\mathsf{S}(x)$ and $\mathsf{W}(x)$ using $O(\log n)$ bits. Let us assume that $\mathsf{F}_b(x) \in \mathbb{R}^{n \times n^2}$ whose $j_0$-th column $\mathsf{F}_b(x)_{j_0} = \mathsf{S}(x)_{j_0} \mathsf{S}(x)_{j_0}^\top \mathsf{W}(x)_{j_0}$ for each $j_0 \in [n]$ (see Definition F.6). Then there are matrices $U_4, V_4, W_4 \in \mathbb{R}^{n \times k_4}$ such that $\|U_4(V_4 \oslash W_4)^\top - \mathsf{F}_b(x)\|_\infty \leq \epsilon/\operatorname{poly}(n)$. We can construct the matrices $U_4, V_4, W_4$ in $n^{1+o(1)}$ time.*

*Proof.* For further simplicity of proofs, we define $\mathsf{R}(x) \in \mathbb{R}^n$ to be a local vector function where $\mathsf{R}(x)_{j_0}$ is $\langle \mathsf{S}(x)_{j_0}, \mathsf{W}(x)_{j_0} \rangle$. We denote the approximation of $\mathsf{R}(x)$ to be $\widetilde{\mathsf{R}}(x)$.

It is noted that a good approximation of $\mathsf{S}(x)_{j_0}$ is $(U_1(V_1 \oslash W_1)^\top)_{j_0,*}^\top$. We denote the approximation of $\mathsf{S}(x)$ to be $\widetilde{\mathsf{S}}(x) = U_1(V_1 \oslash W_1)^\top$.

It is noted that a good approximation of $\mathsf{W}(x)_{j_0}$ is $(U_2(V_2 \oslash W_2)^\top)_{j_0,*}^\top$. Let denote the approximation of $\mathsf{W}(x)$ to be $\widetilde{\mathsf{W}}(x) = U_2(V_2 \oslash W_2)^\top$.

Suppose that $\widetilde{\mathsf{R}}(x)_{j_0} := \langle \widetilde{\mathsf{S}}(x)_{j_0}, \widetilde{\mathsf{W}}(x)_{j_0} \rangle = (U_1(V_1 \oslash W_1)^\top)_{j_0,*} \cdot (U_2(V_2 \oslash W_2)^\top)_{j_0,*}^\top$.

For the side of computation time, we compute $V_1^\top V_2$ first and this takes $n^{1+o(1)}$ time. Then, we compute $W_1^\top W_2$ and this also takes $n^{1+o(1)}$ time.

Next, we have

$$\widetilde{\mathsf{R}}(x)_{j_0} = (U_1(V_1 \oslash W_1)^\top)_{j_0,*} \cdot (U_2(V_2 \oslash W_2)^\top)_{j_0,*}^\top$$

$$= \underbrace{(U_1)_{j_0,*}}_{1 \times k_1} \underbrace{(V_1 \oslash W_1)^\top}_{k_1 \times n^2} \underbrace{(V_2 \oslash W_2)}_{n^2 \times k_2} \underbrace{((U_2)_{j_0,*})^\top}_{k_2 \times 1}$$

$$= \underbrace{(U_1)_{j_0,*}}_{1 \times k_1} \underbrace{((V_1^\top V_2)}_{k_1 \times k_2} \circ \underbrace{(W_1^\top W_2))}_{k_1 \times k_2} \underbrace{((U_2)_{j_0,*})^\top}_{k_2 \times 1}$$

where the first step follows from the definition of $R(x)$, the second step follows from $(AB)_{j_0,*} = e_{j_0}(AB) = (e_{j_0}A)B = A_{j_0,*}B$ for any matrices $A$ and $B$, and the third step is due to Lemma C.13.

Once we have pre-computed $V_1^\top V_2 \in \mathbb{R}^{k_1 \times k_2}$ and $W_1^\top W_2 \in \mathbb{R}^{k_1 \times k_2}$, the above step only takes $O(k_1 k_2)$ time. Since there $n$ coordinates, so the overall time complexity is still $O(nk_1 k_2) = n^{1+o(1)}$.

We can use $\widetilde{S}(x)$ and $\widetilde{R}(x)$ to approximate $F_b(x)$. Let $\widetilde{F}_b(x) = \underbrace{\mathrm{diag}(\widetilde{R}(x))}_{n \times n} \underbrace{\widetilde{S}(x)}_{n \times n^2}$. Because $\mathrm{diag}(\widetilde{R}(x))$ is a diagonal matrix and $\widetilde{S}(x)$ has low-rank representation, then obviously we know how to construct $U_4, V_4, W_4$. Basically $U_4 = \mathrm{diag}(\widetilde{R}(x))U_1$ and $V_4 = V_1$, $W_4 = W_1$.

Now, we need to control the error, and we have

$$\|U_4(V_4 \oslash W_4)^\top - F_b(x)\|_\infty$$
$$= \|\widetilde{F}_b(x) - F_b(x)\|_\infty$$
$$= \max_{j_0 \in [n]} \|\widetilde{S}(x)_{j_0}\widetilde{R}(x)_{j_0} - S(x)_{j_0}R(x)_{j_0}\|_\infty$$
$$= \max_{j_0 \in [n]} \|\widetilde{S}(x)_{j_0}\widetilde{R}(x)_{j_0} - \widetilde{S}(x)_{j_0}R(x)_{j_0} + \widetilde{S}(x)_{j_0}R(x)_{j_0} - S(x)_{j_0}R(x)_{j_0}\|_\infty$$
$$\leq \max_{j_0 \in [n]} \|\widetilde{S}(x)_{j_0}\widetilde{R}(x)_{j_0} - \widetilde{S}(x)_{j_0}R(x)_{j_0}\|_\infty + \|\widetilde{S}(x)_{j_0}R(x)_{j_0} - S(x)_{j_0}R(x)_{j_0}\|_\infty$$

where the first step is due to the definition of $\widetilde{F}_b(x)$, the second step follows from the definition of $F_b(x)$ and $\widetilde{F}_b(x)$, the third step follows from simple algebra, and the last step follows from triangle inequality.

For the 1st term, we have

$$\max_{j_0 \in [n]} \|\widetilde{S}(x)_{j_0}\widetilde{R}(x)_{j_0} - \widetilde{S}(x)_{j_0}R(x)_{j_0}\|_\infty \leq \max_{j_0 \in [n]} \|\widetilde{S}(x)_{j_0}\|_\infty \cdot |\widetilde{R}(x)_{j_0} - R(x)_{j_0}|$$
$$\leq \epsilon/\mathrm{poly}(n)$$

For the 2nd term, we have

$$\max_{j_0 \in [n]} \|\widetilde{S}(x)_{j_0}R(x)_{j_0} - S(x)_{j_0}R(x)_{j_0}\|_\infty \leq \max_{j_0 \in [n]} \|\widetilde{S}(x)_{j_0} - S(x)_{j_0}\|_\infty \cdot |R(x)_{j_0}|$$
$$\leq \epsilon/\mathrm{poly}(n)$$

We complete the proof, by using all three equations we derived above. $\qquad\square$

### F.6 Gradient computation in almost linear time by low rank tensor approximation

We now present our main result regarding the time complexity of our algorithm.

**Theorem F.8** (Main result for fast gradient computation, formal version of Theorem 4.2)**.** *Assuming the entries of $A_1, A_2, A_3, A_4, A_5, E \in \mathbb{R}^{n \times d}$ and $X_1, X_2, X_3, Y_1, Y_2 \in \mathbb{R}^{d \times d}$ are represented using $O(\log n)$ bits. Then, there exist an algorithm that runs in $n^{1+o(1)}$ time to solve $\mathsf{ATAttLGC}(n, d = O(\log n), B = o(\sqrt[3]{\log n}), \epsilon = 1/\mathrm{poly}(n))$ (see Definition 2.9), i.e., our algorithm computes a gradient matrix $\widetilde{g} \in \mathbb{R}^{d \times d^2}$ satisfying $\|\frac{\mathrm{dLoss}(X)}{\mathrm{d}X} - \widetilde{g}\|_\infty \leq 1/\mathrm{poly}(n)$.*

*Proof of Theorem 4.2.* Given size $n \times n^2$ matrices $F(x)$ (see Definition D.6), $F_a(x)$ (see Lemma F.7) and $F_b(x)$ (see Lemma F.5), obviously we know

$$F(x) = F_a(x) - F_b(x).$$

By applying Lemma F.1, Lemma F.2, and Lemma F.3, we confirm that the assumptions in Lemma F.5 and Lemma F.7 hold true. Therefore, we can utilize Lemma F.5 and Lemma F.7 to conclude that

- Let $k_3 = n^{o(1)}$. We know that $\mathsf{F}_a(x)$ has approximate low rank representation $U_3, V_3, W_3 \in \mathbb{R}^{n \times k_3}$, let $\widetilde{\mathsf{F}}_a(x)$ denote $U_3(V_3 \oslash W_3)^\top$.

- Let $k_4 = n^{o(1)}$. We know that $\mathsf{F}_b(x)$ has approximate low rank representation $U_4, V_4, W_4 \in \mathbb{R}^{n \times k_4}$, let $\widetilde{\mathsf{F}}_b(x)$ denote $U_4(V_4 \oslash W_4)^\top$.

- Let $U_5, V_5, W_5 \in \mathbb{R}^{n \times k_5}$ denote the approximate low rank representation for $\mathsf{F}(x)$, call it $\widetilde{\mathsf{F}}(x) = U_5(V_5 \oslash W_5)^\top$. We have $k_5 \leq k_3 + k_4 = n^{o(1)}$.

Thus, Lemmas F.1, F.2, F.3, F.5 and F.7 all are taking $n^{1+o(1)}$ time to compute.

From the Lemma E.6, we know that

$$\frac{\mathrm{dLoss}(x)}{\mathrm{d}x} = \mathrm{vec}(A_1^\top \mathsf{F}(x)(A_2 \otimes A_3))$$

We use $\mathrm{vec}(A_1^\top \widetilde{\mathsf{F}}(x)(A_2 \otimes A_3))$ to do approximation, then

$$\mathrm{vec}(\underbrace{A_1^\top}_{d \times n} \underbrace{\widetilde{\mathsf{F}}(x)}_{n \times n^2} \underbrace{(A_2 \otimes A_3)}_{n^2 \times d^2}) = \mathrm{vec}(\underbrace{A_1^\top}_{d \times n} \underbrace{\widetilde{\mathsf{F}}(x)}_{n \times n^2} \underbrace{(A_2^\top \otimes A_3^\top)^\top}_{n^2 \times d^2})$$

$$= \mathrm{vec}(\underbrace{[U_5 \odot V_5 \odot W_5]}_{n \times n \times n}(A_1^\top, A_2^\top, A_3^\top))$$

$$= \mathrm{vec}(((A_1^\top U_5) \odot (A_2^\top V_5) \odot (A_3^\top W_5))),$$

where the first step is due to Fact C.4, the second step is because of Claim C.20 and Fact C.11, and the last step follows Fact C.12.

The above computation takes $n^{1+o(1)}d + d^3 n^{o(1)}$ time. So, overall time complexity is still $n^{1+o(1)}$.

Recall that $\widetilde{g} \in \mathbb{R}^{d \times d^2}$ and $\frac{\mathrm{dLoss}(X)}{\mathrm{d}X} \in \mathbb{R}^{d \times d^2}$.

We have

$$\|\frac{\mathrm{dLoss}(X)}{\mathrm{d}X} - \widetilde{g}\|_\infty = \|\mathrm{vec}(A_1^\top \mathsf{F}(x)(A_2 \otimes A_3)) - \mathrm{vec}(A_1^\top \widetilde{\mathsf{F}}(x)(A_2 \otimes A_3))\|_\infty$$

$$= \|A_1^\top \mathsf{F}(x)(A_2 \otimes A_3) - A_1^\top \widetilde{\mathsf{F}}(x)(A_2 \otimes A_3)\|_\infty$$

$$= \|A_1^\top (\mathsf{F}_a(x) - \mathsf{F}_b(x))(A_2 \otimes A_3) - A_1^\top (\widetilde{\mathsf{F}}_a(x) - \widetilde{\mathsf{F}}_b(x))(A_2 \otimes A_3)\|_\infty$$

$$\leq \|A_1^\top (\mathsf{F}_a(x) - \widetilde{\mathsf{F}}_a(x))(A_2 \otimes A_3)\|_\infty + \|A_1^\top (\mathsf{F}_b(x) - \widetilde{\mathsf{F}}_b(x))(A_2 \otimes A_3)\|_\infty$$

$$\leq \|A_1\|_\infty \|A_2\|_\infty \|A_3\|_\infty \cdot n^3 \cdot (\|\mathsf{F}_a(x) - \widetilde{\mathsf{F}}_a(x)\|_\infty + \|\mathsf{F}_b(x) - \widetilde{\mathsf{F}}_b(x)\|_\infty)$$

$$\leq \epsilon / \mathrm{poly}(n)$$

where the 1st step is due to definition of $\frac{\mathrm{dLoss}(X)}{\mathrm{d}X}$ in the above, the 2nd step follows from the definition of $\mathrm{vec}(\cdot)$, the 3rd step follows from simple algebra, the 4th step follows from triangle inequality, the 5th step follows from $\|\mathsf{T}(A_1, A_2, A_3)\|_\infty \leq n^3 \cdot \|\mathsf{T}\|_\infty \cdot \|A_1\|_\infty \cdot \|A_2\|_\infty \cdot \|A_3\|_\infty$, where $\mathsf{T}$ is a tensor, and the last step follows from entries in $A_1, A_2, A_3$ are bounded, and $\|\mathsf{F}_a(x) - \widetilde{\mathsf{F}}_a(x)\|_\infty \leq \epsilon / \mathrm{poly}(n)$, $\|\mathsf{F}_b(x) - \widetilde{\mathsf{F}}_b(x)\|_\infty \leq \epsilon / \mathrm{poly}(n)$.

By picking $\epsilon = 1 / \mathrm{poly}(n)$, we complete the proof. $\square$

## G  HARDNESS

In this section, we will show the hardness of our algorithm. In Section G.1, we provide some useful tools for our results. In Section G.2, we present our main hardness results.

### G.1  TOOLS FOR BACKWARD COMPLEXITY

Next, we demonstrate that the tensor attention optimization problem (see Definition 2.8) exhibits favorable behavior when applied to matrices constrained as described in Lemma 5.2:

**Lemma G.1.** *Suppose that a fixed matrix $H \in \mathbb{R}^{n \times n^2}$ with entries in the interval $[1, B_a]$ satisfying that more than half entries of $H$ in each row are equal to $B_a$. Let a matrix $V \in \mathbb{R}^{n^2 \times d}$ with entries in $\{0, 1\}$. For $\lambda \in \mathbb{R}$, let us define $M_\lambda := \exp(\lambda H) \in \mathbb{R}^{n \times n^2}$. We denote the function $f : \mathbb{R} \to \mathbb{R}$ as*

$$f(\lambda) := \| \underbrace{\mathrm{diag}(M_\lambda \mathbf{1}_{n^2})^{-1}}_{n \times n} \underbrace{M_\lambda}_{n \times n^2} \underbrace{V}_{n^2 \times d} \|_F^2,$$

*Then, for every $\lambda \in \mathbb{R}$ we get*

- *$|f'(\lambda)| \leq O(B_a n d)$,*

- *$|f''(\lambda)| \leq O(B_a^2 n d)$.*

*Proof.* Let $G$ denote the $n \times n^2$ matrix $G = \mathrm{diag}(M_\lambda \mathbf{1}_n)^{-1} M_\lambda$. For $i \in [n], j \in [n^2]$, we calculate that $M_{\lambda i,j} = e^{\lambda H_{i,j}}$ and so

$$G_{i,j} = \frac{e^{\lambda H_{i,j}}}{\sum_{k=1}^{n^2} e^{\lambda H_{i,k}}}.$$

For $\ell \in [d]$, let $S_\ell \subseteq [n^2]$ represent the set of 1s in column $\ell$ of $V$, defined as $S_\ell = \{j \in [n^2] \mid V_{j,\ell} = 1\}$. Therefore, for each $i \in [n], \ell \in [d]$, the $(i, \ell)$ entry of the matrix $\mathrm{diag}(M_\lambda \mathbf{1}_n)^{-1} M_\lambda V$ can be shown that

$$(\mathrm{diag}(M_\lambda \mathbf{1}_n)^{-1} M_\lambda V)_{i,\ell} = (GV)_{i,\ell}$$

$$= \sum_{j=1}^{n^2} G_{i,j} V_{j,\ell}$$

$$= \sum_{j \in S_\ell} G_{i,j}$$

$$= \frac{\sum_{j \in S_\ell} e^{\lambda H_{i,j}}}{\sum_{k=1}^{n^2} e^{\lambda H_{i,k}}}.$$

where the 1st step comes from definition, the 2nd step is due to simple algebra, the 3rd step is because of definition of $S_\ell$, and the last step comes from definition of $G$.

Thus, we obtain:

$$f(\lambda) = \sum_{i=1}^{n} \frac{\sum_{\ell=1}^{d} \left( \sum_{j \in S_\ell} e^{\lambda H_{i,j}} \right)^2}{\left( \sum_{k=1}^{n^2} e^{\lambda H_{i,k}} \right)^2}$$

$$= \sum_{i=1}^{n} \frac{\sum_{\ell=1}^{d} \sum_{j_1 \in S_\ell} \sum_{j_2 \in S_\ell} e^{\lambda(H_{i,j_1} + H_{i,j_2})}}{\sum_{k_1=1}^{n^2} \sum_{k_2=1}^{n^2} e^{\lambda(H_{i,k_1} + H_{i,k_2})}}.$$

We define

$$g(\lambda, i) := \sum_{\ell=1}^{d} \sum_{j_1 \in S_\ell} \sum_{j_2 \in S_\ell} e^{\lambda(H_{i,j_1} + H_{i,j_2})}.$$

We also define

$$h(\lambda, i) := \sum_{k_1=1}^{n^2} \sum_{k_2=1}^{n^2} e^{\lambda(H_{i,k_1} + H_{i,k_2})}.$$

By the previous three equations, we have:

$$f(\lambda) = \sum_{i=1}^{n} g(\lambda, i)/h(\lambda, i).$$

As at least half of the entries in each row of $H$ are equal to $B_a$ and all entries lie within the interval $[1, B_a]$, we can bound:

$$\left(\frac{n^2}{2}\right)^2 \cdot e^{2B_a\lambda} \le h(\lambda, i) \le (n)^4 \cdot e^{2B_a\lambda}. \tag{8}$$

Furthermore, since the derivative of $e^{\lambda(H_{i,k_1}+H_{i,k_2})}$ with respect to $\lambda$ is $(H_{i,k_1} + H_{i,k_2}) \cdot e^{\lambda(H_{i,k_1}+H_{i,k_2})}$, we can bound

$$2 \cdot h(\lambda, i) \le \frac{\mathrm{d}h(\lambda, i)}{\mathrm{d}\lambda} \le 2B_a \cdot h(\lambda, i). \tag{9}$$

We may similarly bound

$$0 \le g(\lambda, i) \le d \cdot n^4 \cdot e^{2B_a\lambda}, \tag{10}$$

and

$$2 \cdot g(\lambda, i) \le \frac{\mathrm{d}g(\lambda, i)}{\mathrm{d}\lambda} \le 2B_a \cdot g(\lambda, i). \tag{11}$$

The derivative of $f$ can be bounded by (where the $'$ notation denotes the derivative w.r.t. $\lambda$):

$$\begin{aligned}
f'(\lambda) &= \sum_{i=1}^{n} \frac{g'(\lambda, i) \cdot h(\lambda, i) - g(\lambda, i) \cdot h'(\lambda, i)}{(h(\lambda, i))^2} \\
&\le \sum_{i=1}^{n} \frac{g'(\lambda, i) \cdot h(\lambda, i)}{(h(\lambda, i))^2} \\
&= \sum_{i=1}^{n} \frac{g'(\lambda, i)}{h(\lambda, i)} \\
&\le \sum_{i=1}^{n} \frac{2B_a d \cdot n^4 e^{2B_a\lambda}}{(n^2/2)^2 \cdot e^{2B_a\lambda}} \\
&= \sum_{i=1}^{n} 8B_a d \\
&= 8B_a \cdot nd.
\end{aligned}$$

where the first step is due to the calculation of derivative, the second step is due to basic algebra, the third step is because of cancelling $h(\lambda, i)$, the fourth step is by Eq. (8) ($h(\lambda, i)$ term) and Eq. (11) ($g'(\lambda, i)$ term), the fifth step is due to basic algebra, and the last step is due to basic algebra.

In a similar manner, a lower bound for $f'(\lambda)$ can be,

$$\begin{aligned}
f'(\lambda) &= \sum_{i=1}^{n} \frac{g'(\lambda, i) \cdot h(\lambda, i) - g(\lambda, i) \cdot h'(\lambda, i)}{(h(\lambda, i))^2} \\
&\ge -\sum_{i=1}^{n} \frac{g(\lambda, i) \cdot h'(\lambda, i)}{(h(\lambda, i))^2} \\
&\ge -\sum_{i=1}^{n} \frac{(dn^4 \cdot e^{2B_a\lambda}) \cdot (2B_a \cdot h(\lambda, i))}{((n^2/2)^2 \cdot e^{2B_a\lambda}) \cdot (h(\lambda, i))} \\
&= -\sum_{i=1}^{n} 8B_a d
\end{aligned}$$

$$= -8B_a \cdot nd.$$

where the first step is due to the definition, the second step is due to basic algebra, the third step comes from Eq. (8) ($h(\lambda, i)$ term), Eq. (9) ($h'(\lambda, i)$ term), and Eq. (10) ($g(\lambda, i)$ term), the fourth step is due to basic algebra, and the final step comes from basic algebra.

Finally, we let $f(\lambda, i) := \frac{g(\lambda,i)}{h(\lambda,i)}$, and we can have $f''(\lambda)$ is equal to the following using the quotient rule:

$$\sum_{i=1}^{n} \frac{g''(\lambda, i) - h''(\lambda, i) \cdot f(\lambda, i) - 2 \cdot h'(\lambda, i) \cdot f'(\lambda, i)}{h(\lambda, i)},$$

which we can likewise bound in magnitude by $O(B_a^2 nd)$. $\qquad\square$

We have the following tool from previous work.

**Lemma G.2** (Lemma 5.4 in (Alman & Song, 2024a)). *Suppose that $f : [0, 1] \to \mathbb{R}$ is a twice-differentiable function that satisfy $|f''(\lambda)| \leq b$ for all $\lambda \in [0, 1]$. And for any positive integer $t$, we define*

$$s_t := \sum_{i=0}^{t-1} \frac{f'(i/t)}{t}$$

*Then, we have*

$$|s_t - (f(1) - f(0))| \leq b/t.$$

### G.2 MAIN RESULT FOR LOWER BOUND

Finally, we are prepared to present our main result:

**Theorem G.3** (Main result for hardness, formal of Theorem 5.3). *Let $\gamma : \mathbb{N} \to \mathbb{N}$ be any function with $\gamma(n) = o(\log n)$ and $\gamma(n) = \omega(1)$. Assuming SETH, for any constant $\delta > 0$, it is impossible to solve ATAttLGC($n, d = \Theta(\log n), B = \Theta(\sqrt[3]{\gamma(n) \cdot \log n}), \epsilon = O(1/(\log n)^4)$) (Definition 2.9) in time $O(n^{3-\delta})$ when $E = 0$, $\mathsf{Y} = \mathsf{I}_d$, $\mathsf{X} = \lambda \mathsf{I}_d$ for some scalar $\lambda \in [0, 1]$.*

*Proof of Theorem 5.3.* Let us assume that such an algorithm do exist. Then we can call it $O((\log n)^{11})$ times to refute Lemma 5.2 using parameter $\gamma = \gamma(n)$, i.e., we can get $f(1)$ by solving ATAttLGC with $O((\log n)^{11})$ times.

Suppose that $\mathsf{I}_d \in \mathbb{R}^{d \times d \times d}$ is an identity tensor. Also suppose that the input matrices to Lemma 5.2 are $Q, K_1, K_2, V_1, V_2$. And we set $A_1 = Q$, $A_2 = K_1, A_3 = K_2$, $A_4 = V_1, A_5 = V_2$, $Y = I$, and $X = \lambda \cdot \underbrace{\mathrm{mat}(\mathsf{I}_d)}_{d \times d^2}$, with some $\lambda \in [0, 1]$. Let $f : [0, 1] \to \mathbb{R}$ be defined in Lemma G.1 where $H$ is the matrix $A_1(A_2 \oslash A_3)^\top$, so that $M_\lambda$ is the matrix $\exp(A_1 X (A_2 \otimes A_3)^\top)$ by Fact C.19. It follows from Lemma G.1 and $d = \Theta(\log n)$ that

$$|f''(\lambda)| \leq O(n \log^5 n \cdot (\gamma(n))^2),$$

where $B_a = O(\gamma(n) \log^2 n)$ in Lemma G.1 by the second bullet point of Lemma 5.2.

It is worth noting that $f(0)$ can be computed in $\widetilde{O}(n)$ time because of the all-1s matrix $M_f$. Our final target is to calculate $f(1)$.

From Lemma G.2, $f'(\lambda)$ can be computed on $O(\log^9(n)(\gamma(n))^2) = O(\log^{11} n)$ points up to error $O(1/(\log n)^4)$, and give back their average. Because we have already chosen $X = \lambda I$, $f'(\lambda)$ can be calculated from the gradient $\frac{\mathrm{dLoss}(X)}{\mathrm{d}X}$ in (see Definition 2.9), by our assumed approximated algorithm. $\qquad\square$

## LLM USAGE DISCLOSURE

LLMs were used only to polish language, such as grammar and wording. These models did not contribute to idea creation or writing, and the authors take full responsibility for this paper's content.

