# OpenReview forum: "Training Tensor Attention Efficiently: From Cubic to Almost Linear Time"
_ICLR.cc/2026/Conference — Submitted to ICLR 2026_

### Official Review · Reviewer_pt6n · 2025-10-29

**Soundness:** 3
**Presentation:** 3
**Contribution:** 2
**Rating:** 4
**Confidence:** 3

**Summary:**

In this paper, the authors study the backward pass of tensor attentions, which usually requires $O(n^3)$ computations,
where $n$ is the sequence length. They show under that, certain conditions on the embedding size and size of the entries,
approximately computing the gradient can be done in $n^{1+o(1)}$ time. In addition, they show that without those
assumptions, approximately computing the gradient cannot be done in sub-cubic time.

**Strengths:**

* Overall, the paper is well-written and easy-to-follow.
* Though the algorithm is based on the polynomial approximation method developed in (Alman & Song, 2024), it is not a
  direct application of the same analysis, due to some difference between the forward and backward passes.
* In terms of the lower bounds, while I do not think worst case complexity is the most relevant measure in this setting,
  having negative results is always a good thing and helps justify the assumptions.

**Weaknesses:**

The main weakness is the scope. This paper targets a very specific problem (approximate computation of the
gradient of tensor attentions) in a restricted setting. It is unclear how relevant it is to either practice or theory.
* *Empirical relevance.* The authors require the embedding size $d$ to be $O(\log n)$, which is much smaller than what
  is used in practice. In addition, the runtime complexity appears to heavily depend on $d$. If I understand the method
  correctly, the runtime becomes $\mathrm{poly}(n)$ for some large $\mathrm{poly}$ if $d = \log^C n$, and becomes
  super-polynomial if $d = n^{0.1}$.
* *Theoretical relevance.* Tensor attention was introduced in (Sanford et al., 2023) to solve a very specific task
  (Match3), mainly to demonstrate that, while Match3 might be hard for transformers, it is not inherently
  hard, as there is a variant of transformers that can easily solves this task. I do not see how tensor attentions
  (and approximate computation of its gradient) can be connected to other parts of deep learning theory. In particular,
  it seems that having multiple layers is sufficient to handle many triple-wise relations in less adversarial settings.

**Questions:**

See the weakness section for details.
* Is it possible to allow a larger embedding size by strengthening other parts of the assumption? What is trade-off here?
* How can tensor attention (and the approximation computation of its gradient) be connected broader areas of deep learning theory?

---

### Official Review · Reviewer_NKpC · 2025-10-30

**Soundness:** 3
**Presentation:** 2
**Contribution:** 2
**Rating:** 4
**Confidence:** 2

**Summary:**

The authors theoretically demonstrate that tensor-attention gradients can be computed in almost linear time, breaking the cubic barrier for third-order interactions. While the algorithm lays a solid foundation for higher-order Transformers, large-scale real-world validation is still missing.

**Strengths:**

1. The paper introduces the first sub-cubic gradient algorithm for tensor attention, achieved through a clever combination of polynomial approximation and Kronecker-structured computation, representing a clear theoretical breakthrough.
2. The inclusion of a SETH-based hardness analysis convincingly shows that the bounded-entry assumption is necessary for computational tractability.
3. The proposed formulation is generally applicable and compatible with a wide range of tri-modal attention mechanisms and existing Transformer frameworks.

**Weaknesses:**

1. The paper does not include experiments on real multimodal datasets, leaving the actual acceleration and performance improvements unverified.
2.The theoretical assumptions, such as requiring input matrix entries to be bounded, may not hold in practical deep learning scenarios, limiting real-world applicability.

**Questions:**

1. It remains unclear whether the proposed tensor-attention algorithm can be efficiently implemented within existing deep learning frameworks and hardware environments, such as PyTorch, TensorFlow, or GPU tensor-core architectures.
2. Evaluation on real-world visual or NLP benchmarks would help clarify the effectiveness of the proposed algorithm and determine how well the theoretical advantages translate into practical improvements.

---

### Official Review · Reviewer_1BVG · 2025-10-31

**Soundness:** 2
**Presentation:** 2
**Contribution:** 2
**Rating:** 4
**Confidence:** 3

**Summary:**

This paper derives an efficient algorithm to approximate the gradients of high-order transformers. Such models extend the classical self attention mechanism, where only pairwise interactions are modelled, to higher order interactions where e.g. each token can attend to every possible pairs of tokens in the sentence. This model can also be used to model interactions between different modalities or views of a same object.

The attention model itself was already introduced previously in 2023 and 2024, along with an efficient algorithm to approximate the forward pass computation. The contributions of the submission is to design an analogous efficient algorithm for fast approximation of the backward pass computation.

The main theoretical result shows that the proposed approximation algorithm can achieve epsilon = 1 / poly(n) approximation guarantee in almost linear time to compute gradients over a sequence of length n. The authors also provide a hardness analysis showing that their assumption are tight.

The contribution is only theoretical, no experiments are provided.

**Strengths:**

- The problem of efficient computation of higher-order attention mechanism is relevant

- A hardness analysis is provided to strengthen the result.

- The paper has gained in clarity compared to the version submitted last year at ICLR (that I also reviewed)

**Weaknesses:**

- No experiments are provided to demonstrate the effectiveness (and correctness) of the proposed analysis / algorithm. For such a contribution, focusing on making training of tensor based model learning tractable, experimental validation showcasing the effectiveness of the proposed approach and demonstrating its superiority quantitatively (runtime, memory consumption), even on synthetic data with small models, is required.

- Despite the improvement in clarity compared to submission at ICLR last year, it is very dense and could benefit from discussing more the high level intuition and trying to simplify the exposition of the results.

**Questions:**

- Lemma 3.1: it should be clarified that F(x) is a function of many other variables than X.

- Def 2.8: Replace "as 0.5 ||......" with "where Loss(X) = 0.5 ||..."

- The tensor trick (Fact 4.6) is a well know property of the vec operator (https://en.wikipedia.org/wiki/Vectorization_(mathematics)) and is not used in the main paper as far as I see, could be deferred to the appendix.

---

### Official Review · Reviewer_iaKt · 2025-11-03

**Soundness:** 3
**Presentation:** 3
**Contribution:** 3
**Rating:** 4
**Confidence:** 3

**Summary:**

This paper shows that the backward gradient of tensor attention can be computed in almost linear time, matching the forward computation under a bounded entries assumption. It derives closed-form solution for the gradient, introduces a fast algorithm based on polynomial approximations, and provides hardness proofs establishing necessity and tightness of bounded entries assumption

**Strengths:**

- the paper extends previous work on forward tensor attention computation to the backward pass, completing the picture for efficient attention training.

- the result is important for making tensor attention practical for training, a valuable matter for multi-modal learning and capturing high order interactions

- The tensor operation techniques introduced are non trivial and elegant, leading to crucial improvements for the overall complexity

- The assumptions are demonstrated to be tight; slightly weakening bounded entries assumption can render the improvements unobtainable

- where utilized, visual aids can be helpful but are limited to basic concepts.

**Weaknesses:**

- no empirical validation at all. This is a significant limitation, even for a more theoretical paper for ICLR

- presentation is dense, and some intuitive explanation (e.g., for algorithm) could be useful. There is a long appendix with critical content.

- evidence on practicality of restrictive assumption

- although extension is mentioned, this approach addresses only 3rd order tensors

- no discussion on implementation matters (memory, numerical stabilitry), and quality of polynomial approximation

**Questions:**

see weaknesses above.

---

### Meta-Review · Area_Chair_CXje · 2026-01-08

**Summary:**

There is no empirical experiments provided to validate their proposed approach.

**Reviewer Concerns:**

There is no empirical experiments provided to validate their proposed approach.

**Reviewer Scores:**

No reviewer will increase their score since no rebuttal is provided.

---

### Decision · Program_Chairs · 2026-01-26

Reject